# Characterization of biogenic primary and secondary
# organic aerosols in the marine atmosphere over the East
# China Sea
**Mingjie Kang[1,2,3], Pingqing Fu[1,2,4], Kimitaka Kawamura[5], Fan Yang[2], Hongliang**
**Zhang[6], Zhengchen Zang[7], Hong Ren[2,8], Lujie Ren[1], Ye Zhao[3], Yele Sun[2,8], and**
**Zifa Wang[2,8]**
[1] Institute of Surface-Earth System Science, Tianjin University, Tianjin 300072, China
[2] State Key Laboratory of Atmospheric Boundary Layer Physics and Atmospheric Chemistry,
Institute of Atmospheric Physics, Chinese Academy of Sciences, Beijing 100029, China
[3] State Key Laboratory of Water Environment Simulation, School of Environment, Beijing
Normal University, Beijing 100875, China
[4] Collaborative Innovation Center on Forecast and Evaluation of Meteorological Disasters,
Nanjing University of Information Science & Technology, Nanjing, 210044, China
[5] Chubu Institute for Advanced Studies, Chubu University, Kasugai 487-8501, Japan
[6] Department of Civil and Environmental Engineering, Louisiana State University, Baton
Rouge, Louisiana 70803, USA
[7] Department of Oceanography and Coastal Sciences, Louisiana State University, Baton
Rouge, LA 70803, USA
[8] College of Earth and Planetary Sciences, University of Chinese Academy of Sciences,
Beijing 100049, China
**Correspondence:** Pingqing Fu (fupingqing@tju.edu.cn)

## Abstract

Molecular composition and abundance of sugars and secondary organic aerosols (SOA) from biogenic sources over the East China Sea were investigated based on gas chromatography/mass spectrometry. Biogenic SOA tracers and sugars exhibit higher levels in the samples affected by continental air masses, demonstrating the terrestrial outflows of organic matter to the East China Sea. Glucose was the dominant sugar species (0.31–209 ng $m^{-3}$, 18.8 ng $m^{-3}$), followed by mannitol – a fungal spore tracer. All sugar compounds show generally higher average concentrations in the nighttime than in the daytime. 3-Methyl-1,2,3-butanetricarboxylic acid, one higher-generation photooxidation tracer of monoterpene SOA, was found to be the most abundant species among measured biogenic SOA markers, suggesting the input of aged organic aerosols through long-range transport. Fungal-spore-derived organic carbon (OC) was the biggest contributor to total OC (0.03–19.8%, 3.1%), followed by sesquiterpene-derived secondary OC (SOC), biomass-burning-derived OC, monoterpene- and isoprene-derived SOC. Larger carbon percentages of biogenic primary OCs and SOCs in total OC presented in the terrestrially influenced aerosols indicate significant contributions of continental aerosols through long-range transport. Positive matrix factorization results illustrate that the secondary nitrate and biogenic SOA, biomass burning, and fungal spores were the main sources of OC in marine aerosols over the East China Sea, again highlighting the importance of Asian continent as a natural emitter of biogenic organic aerosols together with anthropogenic aerosols over the coastal marine atmosphere.

## 1   Introduction

Oceans cover more than 70% of the Earth's surface and marine aerosols contribute significantly to the global aerosol load (O'Dowd et al., 2004), playing an important role in the albedo, atmospheric processes, atmospheric chemistry, climate, and biogeochemical cycling of nutrients (O'Dowd and de Leeuw, 2007; Shi et al., 2011). Such influences depend on the chemical composition and concentrations of marine aerosols. In recent years, significant abundances of organic matter in marine aerosol and their importance to the cloud condensation nuclei (CCN) formation as well as their direct and indirect radiative effects have been reported (Tervahattu et al., 2002; O'Dowd et al., 2004; Facchini et al., 2008; Ovadnevaite et al., 2011a; Ovadnevaite et al., 2011b; Pringle et al., 2010; Bougiatioti et al.,

2011; Sciare et al., 2009). However, information about marine organic aerosol remains poor
owing to various emission sources, complex formation mechanisms, and limited field
measurements regarding their chemical composition and concentrations (O'Dowd and de
Leeuw, 2007; Cavalli et al., 2004). Therefore, it is necessary to investigate the loadings,
molecular compositions and sources of marine organic aerosols, especially in coastal and
offshore regions where the land/ocean interaction is active.
In general, sources of organic compounds in marine aerosols comprise natural and
anthropogenic emissions. For example, surface-active organic matter of biogenic origin in the
ocean (e.g. bacteria, viruses and detritus) can be transferred to the marine atmosphere by
bubble-bursting processes (Gantt and Meskhidze, 2013; Gershey, 1983; Mochida et al., 2002;
Sciare et al., 2009). Terrestrial air masses also bring plentiful organic matter derived from
natural and/or anthropogenic activities to the oceanic atmosphere via long-range atmospheric
transport (Zhou et al., 1990; Uematsu et al., 2010; Kumar et al., 2012; Hawkins et al., 2010;
Srinivas et al., 2011; Kang et al., 2017). Sugars, important water-soluble organic constituents
of atmospheric particulate matter, are substantial in both continental (Pashynska et al., 2002;
Yttri et al., 2007; Fu et al., 2008; Jia and Fraser, 2011; Iinuma et al., 2007; Cong et al., 2015)
and marine aerosols (Simoneit et al., 2004b; Fu et al., 2011). Due to their ubiquity and
abundance, sugars can be used to elucidate sources and transport of atmospheric organic
aerosols. Levoglucosan along with its two isomers, mannosan and galactosan, as the primary
thermal alteration products by the pyrolysis of cellulose and hemicellulose, has been
recognized as specific tracers for biomass burning (Simoneit, 2002). Trehalose is a recognized
fungal carbohydrate, which can be indicative of soil dust (Fu et al., 2012; Feng and Simpson,
2007). Arabitol and mannitol are tracers for airborne fungal spores (Bauer et al., 2008).
Sucrose could serve as a marker for airborne pollen grains (Pacini, 2000; Fu et al., 2012),
while fructose and glucose can be emitted from plant pollen, fruits and detritus (Pacini, 2000;
Speranza et al., 1997; Baker et al., 1998). These sugar compounds have been detected in some
marine aerosols and effectively used as key tracers to assess contributions of different
emission sources (Simoneit et al., 2004b). Despite their importance, the knowledge about
molecular characterization of sugars in marine aerosols is still rare because of various sources
and inconvenience of sampling.
Apart from primary sources, atmospheric volatile organic compounds (VOCs) emitted from
ocean and/or continents also have significant impacts on marine aerosols. VOCs can react
with oxidants, such as ozone ($O_3$), nitrogen oxides ($NO_x$) and OH radicals in the atmosphere,

producing secondary organic aerosols (SOA) (Atkinson and Arey, 2003; Gantt and Meskhidze, 2013; Meskhidze and Nenes, 2006; Hu et al., 2013; Claeys et al., 2004a; Claeys et al., 2007; Jaoui et al., 2005; Hallquist et al., 2009). SOAs contribute substantially to atmospheric particulate OM and strongly affects the regional and global air quality, climate and human health (Zhu et al., 2016; Chen et al., 2017). Recent studies have revealed that SOA is an important or even the dominant contributor to $PM_{2.5}$ during heavy haze events in China (Cao et al., 2017). Due to poor understanding of sources and formation mechanisms of SOA, accurately measuring and modelling SOA concentrations remains a big issue (Zheng et al., 2017). It was reported that emissions of biogenic VOCs (BVOCs) were one order of magnitude larger than those of anthropogenic VOCs on a global scale (Guenther et al., 2006). BVOCs emitted from terrestrial vegetation include isoprene, monoterpenes and sesquiterpenes as well as other compounds (Guenther et al., 2006; Goldstein and Galbally, 2007), which could exert large impacts on marine aerosols through continental aerosols' outflow. On the other hand, marine phytoplankton and seaweeds can release isoprene and other BVOCs as well (Yokouchi et al., 1999; Shaw et al., 2010), especially during the phytoplankton blooms (Hu et al., 2013). Marine-derived SOAs by oxidation of phytoplankton-produced isoprene have been shown to remarkably influence the chemical composition of marine CNN and cloud droplet number, forming an indirect climatic effect (Gantt et al., 2009; Bikkina et al., 2014; Meskhidze and Nenes, 2006). In addition to isoprene, the photooxidation products of oceanic monoterpenes (e.g. α- and β-pinene) were established as well (Fu et al., 2011; Yassaa et al., 2008; Gantt et al., 2009). However, to date, the accurate molecular characterisation and spatial distribution of biogenic SOAs (BSOA) in the marine atmosphere are not well known due to their chemical complexity and process nonlinearity (Chen et al., 2017). Consequently, there remains an urgent need for studies about marine BSOA on a regional and/or global scale currently.

In this study, we investigated molecular compositions, abundances and spatial distributions of sugars and BSOA tracers in marine aerosols over the East China Sea (ECS), which is located between the east coast of Asian continent and the western North Pacific. The ECS is an oceanic region susceptible to the influence from outflow of continental OM from natural and anthropogenic activities in the mainland. The contributions of secondary organic carbon (SOC) from isoprene, monoterpene and sesquiterpene to the total OC were estimated. Positive matrix factorization (PMF) was also used to estimate the contributions of sources identified by biomass burning tracers (e.g. levoglucosan), primary saccharides including fungal spore and

pollen tracers, BSOA tracers from oxidation of BVOCs, inorganic ions and some other
reliable source markers.

## 2   Materials and methods

### 2.1   Aerosol sampling

Marine total suspended particles (TSP) were collected during May 18 to June 12, 2014. The
sampling was performed using a high-volume air sampler (Kimoto, Japan) at an airflow rate
of 0.8 m$^3$ min$^{-1}$ on board of the KEXUE-1 Research Vessel during a National Natural Science
Foundation of China (NSFC) sharing cruise. Figure 1 shows the cruise tracks and
concentrations of chlorophyll-a over the ECS. Detailed sampling information and map of
sampling sites are described elsewhere (Kang et al., 2017).

### 2.2   Organic species analysis

Filter aliquots were ultrasonically extracted for 10 min three times using
dichloromethane/methanol mixture (2:1, v/v). The solvent extracts were filtered through
quartz wool packed in a Pasteur pipette and concentrated with a rotary evaporator, and then
blown down to dryness with pure nitrogen gas. The extracts were reacted with 50 μL of N,O-
bis-(trimethylsilyl)trifluoroacetamide (BSTFA) containing 1% trimethylsilyl chloride and 10
μL of pyridine at 70°C for 3 hour in order to convert COOH and OH to the corresponding
trimethylsilyl (TMS) esters and ethers. After the reaction, 140 μL internal standard (C$_{13}$ *n*-
alkane, 1.43 ng μL$^{-1}$) was added to the derivatives before injection into gas
chromatography/mass spectrometry (GC/MS).

### 2.3   Gas chromatography/mass spectrometry

Two organic species (i.e. sugars and SOA tracers) were determined on an Agilent model 7890
GC coupled to an Agilent model 5975C mass-selective detector (MSD). The GC instrument
was equipped with a split/splitless injector and a DB-5ms fused silica capillary column (30 m
× 0.25 mm i.d., 0.25 μm film thickness) with the GC oven temperature programmed from
50°C (2 min) to 120°C at 15°C min$^{-1}$, and then to 300°C at 5°C min$^{-1}$ with final isotherm
hold at 300°C for 16 min. Helium was used as carrier gas. The GC injector temperature was
maintained at 280°C. The mass spectrometer was operated in the Electron Ionization (EI)
mode at 70 eV and scanned in the *m/z* range of 50 to 650 Da. Data were processed using
ChemStation software. Each compound was determined through comparing mass spectra with
those of authentic standards or literature data. GC/MS response factors were obtained with
authentic standards or surrogate standards. Recoveries of the standards that were spiked onto
pre-combusted quartz filters and measured as the samples (n = 3) were generally better than
80%. A field blank filter was treated as the real samples for quality assurance. The present
data were corrected with field blanks but not for recoveries.
**2.4   Positive matrix factorization (PMF) analysis**
To further investigate the potential sources of marine organic aerosols, the positive matrix
factorization (PMF) analysis are used in this study. For this analysis, the measured ambient
concentrations less than or equal to the method detection limit (MDL) were replaced by
MDL/2 and associated uncertainties were (5/6)*MDL. For the concentrations greater than the
MDL, the calculation of uncertainty is based on the following equation:

$$Uncertainty = \sqrt{(error\ fraction*concentration)^2 + (0.5*MDL)^2}$$

165    (1)

where the error fraction is a user-provided estimation of the analytical uncertainty of the
measured concentration or flux. In the present study, the error fraction was set as 0.2 for all
species for receptor-oriented source apportionment analyses (Han et al., 2017).

# 3   Results and discussion

Concentrations of sugars and biogenic SOA tracers in marine aerosols over the ECS were
presented in Table 1. Total abundance of the quantified sugar compounds ranged from 1.8–
950 ng m$^{-3}$ with an average of 81.5 ng m$^{-3}$ much lower than those in urban atmosphere in
China (Wang et al., 2006), and the nighttime aerosols contained more sugars (average 90.1 ng
m$^{-3}$) than the daytime ones (72.6 ng m$^{-3}$). Individual sugar compounds also showed higher
nighttime concentrations, but still lower than those in urban aerosols (Wang et al., 2011). The
total biogenic SOA tracers were in the range of 1.1–135 ng m$^{-3}$ (average 22.9 ng m$^{-3}$) with
lower nighttime abundance (22.2 ng m$^{-3}$) than daytime (23.6 ng m$^{-3}$). In contrast to
monoterpene and sesquiterpene SOA tracers, the isoprene SOA tracers presented higher levels
in the nighttime aerosols (9.6 ng m$^{-3}$) than those in the daytime samples (7.1 ng m$^{-3}$).
Generally, higher values of these organics were observed in coastal regions compared with
those far away from mainland (Fig. 2, Fig. 4 and Fig. S1), similar to the spatial pattern of
lipids, PAHs and phthalates in our previous report (Kang et al., 2017), suggesting the impact
from the outflow of continental OM on the basis of back trajectory analysis (Fig. S2-S6).
## 3.1 Sugars and sugar-alcohols
Sugars, a major class of water-soluble organic constituents in the atmosphere, have been
reported to be ubiquitous in marine aerosols (Fu et al., 2011; Chen et al., 2013; Simoneit et al.,
2004b; Burshtein et al., 2011). Most of them can serve as tracers of biological primary aerosol
particles and biomass burning (Simoneit, 2002; Bauer et al., 2008). Eleven sugar compounds
including anhydrosugars (levoglucosan, galactosan, mannosan), sugars (fructose, glucose,
sucrose and trehalose) and sugar alcohols (erythritol, arabitol, mannitol and inositol) were
measured in marine aerosols collected over the ECS. The concentrations of sugars are shown
in Table 1 and Fig. 2. Glucose ($0.31–209$ ng m$^{-3}$, mean 18.8 ng m$^{-3}$) and mannitol ($0.03–169$
ng m$^{-3}$, 16.3 ng m$^{-3}$) were the dominant species, followed by sucrose ($0.09–216$ ng m$^{-3}$, 11.7
ng m$^{-3}$), trehalose ($0.08–96.0$ ng m$^{-3}$, 9.4 ng m$^{-3}$), fructose ($0.09–106$ ng m$^{-3}$, 9.1 ng m$^{-3}$), and
levoglucosan ($0.09–64.3$ ng m$^{-3}$, 7.3 ng m$^{-3}$) (Fig. 3). Chen et al. (2013) also reported that
glucose and mannitol are the major sugar compounds detected in marine aerosols collected at
Chichi-Jima Island during spring and summer. Moreover, nighttime concentrations of all
sugars were generally higher than the daytime ones, likely due to lower height of planetary
boundary layer (PBL) and the land breeze carrying plentiful terrestrial OM at night.
Figure 2 presents the temporal variations in sugar compounds determined in the marine
aerosol samples collected over the ECS. The concentrations of total sugars were characterized
by higher levels in the regions close to continent and/or influenced by land air masses,
indicating substantial influence of continental outflow compared to marine air over the
pelagic ocean.
## 3.1.1 Anhydrosugars
Levoglucosan, a thermal degradation production of cellulose (Simoneit, 2002) and a specific
indicator for biomass burning (Simoneit et al., 1999), can largely modify the chemical
composition of atmospheric aerosols on a regional to global scale. Due to its water solubility,
levoglucosan contribute to water-soluble organic carbon in aerosols, significantly enhancing
the hygroscopic properties of atmospheric aerosols (Mochida and Kawamura, 2004). The
abundance of levoglucosan detected in the present study (0.09–41.2 ng m$^{-3}$, mean 6.1 ng m$^{-3}$
and 0.10–64.3 ng m$^{-3}$, mean 8.4 ng m$^{-3}$ for day and night, respectively) was not the highest
among measured sugar species (Fig. 3 and Table 1). The average concentration of
levoglucosan is close to those reported at Gosan, Jeju Island in summer (mean 8.0 ng m$^{-3}$) (Fu
et al., 2012). During the whole sampling period, higher concentrations of levoglucosan were
found in the samples under the effect of continental air masses based on five-day backward
trajectories (Fig. S2-S6), consistent with our previous report about lipids, PAHs and
phthalates (Kang et al., 2017). Mannosan and galactosan, isomers of levoglucosan, are
produced by pyrolysis of cellulose and hemicellulose and can also act as biomass burning
tracers (Fabbri et al., 2009; Simoneit, 2002). Mannosan and galactosan were detected in the
samples with similar variation trends to levoglucosan (Table 1 and Fig. 2), further indicating
the strong impact of continental biomass burning activities on the marine atmosphere. As
shown in Table 1 and Fig. 3, the nighttime average concentrations of all three anhydrosugars
were higher than the daytime ones, probably attributable to enhanced biomass combustion
and lower height of PBL as well as the land breeze during the night.
In addition, hard wood tend to contain higher levels of cellulose than hemicellulose, therefore
the mass concentration ratios of levoglucosan to mannosan (L/M) can be utilized as a
diagnostic parameter for diverse biomass burning substrates (Zhu et al., 2015). Previous
studies found that L/M ratios from softwood are in a range of 3–10 and those from hardwood
are 15–25, while the ratios from crop residues are even higher (25–40) (Zhu et al., 2015). In
this study, the L/M ratios range from 0.71–32 with an average of 4.2, and higher values were
observed in the terrestrially influenced aerosols, especially coastal areas near Fujian province
(Fig. S7). The lower L/M ratios in our study (average 4.2) suggest that the aerosols were
mainly associated with the burning of softwood, consistent with the lower L/M ratios (2.1–4.8)
observed on Okinawa Island in May–June (Zhu et al., 2015) and Mt. Fuji (4.6–7.6, mean 5.5)
(Fu et al., 2014). It is worth noting that a couple of samples near Fujian Province in Southeast
China were characterized by a higher L/M ratio (>20) (Fig. S7), agreeing well with the values
of straw burning smokes (Chen et al., 2013). Zhu et al. (2015) also reported combustion of
agricultural residues, peat and wood could contribute to high L/M ratios. Therefore, these
higher values probably indicate emissions from terrestrial burning of straw residues, which
could affect the chemical components of aerosols in the western North Pacific via long-range
atmospheric transport as confirmed by backward trajectories (Fig. S5).

### 3.1.2 Sugar alcohols

Sugar alcohols detected in these samples consist of arabitol, mannitol, inositol and erythritol, which had similar temporal patterns (Fig. 2e-h). Higher abundances of sugar alcohols observed in the terrestrially influenced aerosol samples suggesting a significant contribution from terrestrial source (Fig. S2-S6). While the marine fungi and algae, which can also release fungal spores into the marine atmosphere via bubble bursting, contribute little to the sugar alcohols in the coastal aerosols. Obviously, mannitol and arabitol were the most abundant sugar alcohols detected in the present study (Fig. 3 and Table 1), ranging from 0.03–169 ng $m^{-3}$ (16.3 ng $m^{-3}$) and 0.02–51.0 ng $m^{-3}$ (5.1 ng $m^{-3}$), respectively. A strong positive correlation between arabitol and mannitol were found in marine aerosols ($p < 0.001$, $r = 0.996$, $N = 51$), suggesting a similar origin. Mannitol and arabitol are very common in fungi, and are the most frequently occurring sugar alcohols in plants (Burshtein et al., 2011), which can be utilized to assess the contribution of fungal spores to the aerosol OC (Bauer et al., 2008). For instance, mannitol is particularly abundant in algae (Burshtein et al., 2011). The peak concentrations of sugar alcohols were found in the nighttime samples, which can be attributed to the increased activities of yeasts and fungi at night (Graham et al., 2003). Moreover, the maxima concentrations of these sugar polyols, especially arabitol and mannitol, were observed in offshore regions in early summer (June) (Fig. 2e-h), likely due to the more active microbial activities resulting from warmer temperature in June and more biota in coastal regions (Fig. 1).

### 3.1.3 Sugars

Glucose, fructose, sucrose, and trehalose are the primary saccharides measured in marine aerosols over the ECS. Glucose was the dominant sugar compound (0.31–209 ng $m^{-3}$, 18.8 ng $m^{-3}$), followed by sucrose (0.09–216 ng $m^{-3}$, 11.7 ng $m^{-3}$) (Fig. 2i-l and Fig. 3). Glucose and fructose originate from plant materials, such as pollen, fruits and their fragments (Fu et al., 2012; Graham et al., 2003). Both glucose and fructose presented higher levels in the coastal areas, demonstrating great contribution from terrestrial vegetation. Medeiros and Simoneit (2007) reported that high abundance of glucose associated with lower molecular weight fatty acids (mainly $C_{16}$) was attributed to the spring bloom of algae. In our study, glucose correlated well with measured $C_{16:0}$ fatty acid ($p = 0.001$, $r = 0.46$, $N = 51$), which mainly emitted from the ocean surface via sea spray (Kang et al., 2017), suggesting that marine sources also contributed to the particulate glucose in the oceanic atmosphere. It was noted that

fructose had a strong correlation with glucose ($p < 0.001$, $r = 0.94$, N = 51), indicating they share similar sources. Sucrose, as a predominant sugar species in the phloem of plants as well as developing flower buds, is reported to be the richest and dominant component of airborne pollen grains (Graham et al., 2003). High abundance of sucrose along with fructose and glucose observed in our study implies a large emission of airborne pollen grains into the marine atmosphere over the ECS. Significantly, the concentration of glucose, fructose and sucrose were highest at the beginning of June in early summer (Fig. 2i-l), probably attributable to an enhanced pollen emission, because pollen counts tend to be highest in late spring/early summer in temperate zones (Graham et al., 2003). Such peak concentrations in early June are in accord with a previous study by Pashynska et al. (2002). Trehalose, a fungal metabolite, is present in a variety of microorganisms (fungi, bacteria, yeast and algae), and a few higher plants as well as invertebrates (Medeiros et al., 2006a). Thus, trehalose can be used as a microbial biomarker and stress protectant. Furthermore, trehalose was reported to be the most abundant sugar in soil (Rogge et al., 2007; Medeiros et al., 2006b; Jia and Fraser, 2011), thus the enrichment of trehalose in aerosols can be indicative of soil resuspension and unpaved road dust (Simoneit et al., 2004a; Fu et al., 2012). The trehalose in our study was positively correlated with non-sea-salt calcium (nss-$Ca^{2+}$) ($p < 0.01$, $r = 0.43$, N = 51), the best tracer for soil dust (Virkkula et al., 2006), suggesting the atmospheric trehalose over the ECS was mainly derived from resuspension of soil particles. Higher abundance of trehalose in coastal regions (Fig. 2l), may be in connection with the outflow of Asian dust primarily occurring in winter/spring (Fu et al., 2012), and inland soil resuspension/dust aerosols.

### 3.2 Secondary organic aerosols

The sum of all SOA tracers ranged from 1.1–135 ng m$^{-3}$ (22.9 ng m$^{-3}$), which were higher than previous report of marine aerosols (0.19–27 ng m$^{-3}$, 6.6 ng m$^{-3}$) (Fu et al., 2011), but much lower than those in the continental sites (Fu et al., 2010; Ding et al., 2014). These differences suggest that the major source of SOA tracers over the ECS is of terrestrial origin. Specifically, the total concentrations of detected isoprene SOA-tracers ranges from 0.15–64.0 ng m$^{-3}$ (8.4 ng m$^{-3}$), comparable to those reported in marine aerosols (mean 8.5 ng m$^{-3}$) (Hu et al., 2013), but lower than those of urban aerosols (Ren et al., 2017; Ding et al., 2014); total monoterpene SOA-tracers in the range of 0.26–87.2 ng m$^{-3}$ (11.6 ng m$^{-3}$) lower than those in urban aerosols as well (Ren et al., 2017); total sesquiterpene SOA-tracers ranged between 0.16 and 17.2 ng m$^{-3}$ (2.9 ng m$^{-3}$). Such phenomena that BSOA derived from monoterpenes

are more abundant than those from isoprene were also reported in Chinese urban areas (Ren et al., 2017). On the whole, biogenic SOA tracers exhibited higher loadings in the coastal areas than those remote sampling sites, further indicating that long-range atmospheric transport of terrestrial aerosols has significant influence on the chemical composition and abundance of SOA over oceans. Marine VOC spatial distributions are expected to be linked to the distributions of photosynthetic pigments in seawater, such as chlorophyll-a (Ooki et al., 2015). The higher concentrations of chlorophyll-a in the coastal regions (Fig. 1) stand for higher biological activities and more emission of VOCs, which agree well with higher SOA tracers over the coastal waters. Such spatial variations in biogenic SOA tracers are in agreement with a previous report about marine organic aerosols collected during a round-the-world cruise (Fu et al., 2011).

### 3.2.1 Isoprene SOA tracers

Isoprene is a reactive biogenic hydrocarbon and primarily originates from terrestrial photosynthetic vegetation (e.g. trees and plants). Marine phytoplankton and seaweed can also emit isoprene (Yokouchi et al., 1999; Shaw et al., 2010). Moreover, bacteria produce isoprene as well, and the bacterial isoprene production is temperature-dependent (Kurihara et al., 2010). In spite of much lower emission strength in the ocean region, more recent researches have suggested that the oceanic source of isoprene significantly impact atmospheric chemistry and cloud microphysical properties in the remote marine boundary layer because of its high reactivity (about 1−2 hour lifetime) (Hackenberg et al., 2017).

Six isoprene SOA tracers, including 2-methylglyceric acid, three $C_5$-alkene triols, and two 2-methyltetrols (2-methylthreitol and 2-methylerythritol), were identified in the marine aerosols over ECS. Isoprene SOA tracers showed diurnal variations with higher average concentrations at night (Fig. 4a-c, Fig. 5, Fig. S8), consistent with the report by Fu et al. (2010). The higher abundance during the night can be explained by the enhanced gas-to-particle partition at cooler temperatures during the nighttime and/or increased input of continent-originated isoprene-SOA into the oceanic atmosphere via land-sea breeze circulations at night. However, T-test showed that the difference between daytime and nighttime concentrations for 2-methyltetrols, $C_5$−alkene triols and 2-MGA was not that significant ($p > 0.05$). As expected, much higher concentrations of isoprene tracers were observed in coastal regions, where continental outflows exert larger effects from spring to early summer. For the remote ocean, terrestrial sources have weak impacts because of the

short atmospheric lifetime of isoprene and the dilution effects during long-range atmospheric
transport. In addition to the effect of continental outflow, more nutrients in the coastal and
estuarine regions could be another factor responsible for the higher levels of isoprene SOA
tracers compared to the pelagic areas. Because nutrient-rich surface water can promote the
development of phytoplankton blooms and increases chlorophyll-a concentrations.
Chlorophyll-a is a measure of phytoplankton, or algal, biomass (Quinn et al., 2014) and
currently most widely used proxy for predicting isoprene concentrations in water
(Hackenberg et al., 2017). Numerous studies reported the positive relationship between
isoprene emission and chlorophyll-a in the surface seawater (Hackenberg et al., 2017; Zhu et
al., 2016). In the present study, the temporal and spatial distributions of chlorophyll-a at the
ECS surface during the whole sampling period are characterized by higher coastal levels, such
as waters near Zhejiang and Fujian provinces, but lower abundance in eastern Taiwan and the
remote sea (Fig. 1). Thus, high chlorophyll-a waters in coastal locations mean more isoprene
emissions than remote open waters. However, the isoprene in the remote ocean may mainly
originate in situ from biological production by marine biota at the ocean surface.
The low-$NO_x$ products 2-methyltetrols with mass concentrations ranging from 0.03–41.9 ng
$m^{-3}$ (4.8 ng $m^{-3}$) were the major species among the isoprene SOA tracers, in line with early
report about summer aerosols in China (Ding et al., 2014). Specially, concentration ranges of
2-methyltetrols were 0.11–17.0 ng $m^{-3}$ (3.8 ng $m^{-3}$) during the daytime and 0.03–41.9 ng $m^{-3}$
(5.7 ng $m^{-3}$) at night with 2-methylerythritol being about 2.1-fold more abundant than 2-
methylthreitol. This ratio is similar to those calculated in previous studies (Fu et al., 2010; Ion
et al., 2005; Cahill et al., 2006). The atmospheric levels of 2-methyltetrols are comparable to
those reported in marine aerosols collected during a round-the-world cruise (0.07–15 ng $m^{-3}$,
2.4 ng $m^{-3}$) (Fu et al., 2011). However, these values are small compared to those from
terrestrial emissions, such as mountain (Fu et al., 2014; Cahill et al., 2006) and forest aerosols
(Miyazaki et al., 2012; Fu et al., 2010; Claeys et al., 2004a). A significant positive
relationship between 2-methyltetrols and levoglucosan ($p < 0.001$, $r = 0.87$, N = 51) in our
study suggests that biomass burning may also generate the precursors of 2-methyltetrols
followed by photochemical reactions (Xie et al., 2014). Besides, 2-methyltetrols correlated
with $C_{29}$ *n*-alkane (the dominant species of terrestrial higher plant waxes) as well ($p < 0.001$, $r$
= 0.74, N = 51), suggesting these organic tracers originate from higher plants or from similar
source regions as well. The correlations above illustrate that terrestrial emissions (e.g.
biomass burning and higher plants) contributed significantly to the levels of 2-methyltetrols in
the marine atmosphere.
$C_5$-Alkene triols are formed from photooxidation of isoprene under low-$NO_x$ conditions
(Surratt et al., 2006; Lin et al., 2013). They were detected in all the samples ranging from
(0.03–14.6 ng m$^{-3}$, 2.2 ng m$^{-3}$), which are higher than those reported in Arctic aerosols (0.01–
0.15 ng m$^{-3}$) (Fu et al., 2009) and other marine aerosols (0.002–4.6 ng m$^{-3}$, 0.65 ng m$^{-3}$) (Fu
et al., 2011); but lower than the atmospheric levels of mountain aerosols (Fu et al., 2014; Fu
et al., 2010) and subtropical urban aerosols from Hong Kong (Hu et al., 2008). Such a
difference illustrates the outflow of continental aerosols, which can be confirmed by the
significant positive correlations between $C_5$-alkene triols and tracers of terrestrial emissions,
such as $C_{29}$ $n$-alkanes ($p < 0.001$, $r = 0.78$, N = 51) and levoglucosan ($p < 0.001$, $r = 0.87$, N =
51). It was found that 2-methyltetrols correlated well with $C_5$−alkene triols in marine aerosols
over ECS ($p < 0.001$, $r = 0.86$, N = 51), suggesting a similar formation mechanism or
common sources. However, the mass concentration ratios of $C_5$-alkene triols to 2-
methyltetrols showed significant variation in different sampling sites (Fig. 6a), indicating
different formation pathways, consistent with a previous report (Fu et al., 2010).
Concentrations of 2-methylglyceric acid (2-MGA), formed by photooxidation of isoprene
under high-$NO_x$ ($NO_x = NO + NO_2$) conditions (Surratt et al., 2006), were in the range of
0.09–8.3 ng m$^{-3}$ (1.4 ng m$^{-3}$) being greatly lower than those in mountain aerosols (Cahill et al.,
2006; Fu et al., 2010; Fu et al., 2014), implying much stronger influence of NOx on isoprene
SOA formation in continental aerosols. 2-MGA is a possible further oxidation product of
methacrolein and methacrylic acid, which are two major gas-phase oxidation products of
isoprene (Claeys et al., 2004b; Edney et al., 2005; Fu et al., 2009). 2-MGA was related to $C_{29}$
$n$-alkanes ($p < 0.001$, $r = 0.84$, N = 51) and levoglucosan ($p < 0.001$, $r = 0.81$, N = 51) as well,
again suggesting a terrestrial input. These isoprene-SOA tracers in marine aerosols over the
ECS may stem from terrestrial higher plants and biomass burning, and then were oxidized
during the transport to the oceanic atmosphere. Similar correlations between isoprene-derived
SOA tracers and levoglucosan were mentioned in previous study, which reported that biomass
burning enhanced the isoprene-SOA formation (Li et al., 2018). Additionally, 2-methyltetrols
are higher generation products than 2-MGA (Fu et al., 2014), but the ratios of 2-MGA to 2-
methyltetrols did not vary significantly with cruise track. However, a couple of high ratios
were observed in the ocean-air influenced aerosols, suggesting the importance of marine
source for fresh isoprene-derived SOAs in the atmosphere (Fig. 6b).
$NO_3^-$ was found to be related to 2-MGA, 2-methyltetrols and $C_5$-alkene triols ($p < 0.001$, $r =$
0.60–0.71, N = 51). The ratio of 2-MGA to 2-methyltetrols was found to be negatively
correlated with particulate $NO_3^-$ ($p < 0.05$, $r = -0.31$, N = 51). These relationships reveal that
there may be a close connection between formation of isoprene SOA and $NO_3^-$ in the marine
boundary layer.
### 3.2.2 Monoterpene SOA tracers
The detected α/β-pinene oxidation products in the study consist of pinonic, pinic acids, 3-
hydroxyglutaric acid (3-HGA) and 3-methyl-1,2,3-butanetricarboxylic acid (MBTCA). They
are derived from the photooxidation of α/β-pinene with $O_3$ and OH radicals (Hoffmann et al.,
1997; Yu et al., 1999; Glasius et al., 2000; Iinuma et al., 2004). Since monoterpenes account
for around 35% of the global biogenic VOCs' emissions, these compounds have been utilized
to estimate the role of monoterpene oxidation in the SOA formation (Griffin et al., 1999).
Monoterpenes were chiefly emitted from needle leaf trees. In this study, monoterpene-SOA
tracers were found to be positively correlated to levoglucosan with $p < 0.001$ ($r$ ranges from
0.68–0.82), indicating terrestrial biomass burning made substantial contributions to the
formation of monoterpenes, then being transported to the oceanic atmosphere. Generally, total
monoterpene-SOA tracers in our study showed a major peak in spring (Fig. S1g), in
agreement with a previous study (Zhu et al., 2016). Pinonic and pinic acids, well-known
tracers for α/β-pinene, ranged from 0.02–1.6 ng m$^{-3}$ (0.47 ng m$^{-3}$) and 0.16–14.9 ng m$^{-3}$ (3.4
ng m$^{-3}$), respectively (Table 1). Their concentrations were higher than those in high Arctic
aerosols (average 69 pg m$^{-3}$ and 514 pg m$^{-3}$, respectively) (Fu et al., 2009). In addition,
concentrations of pinic acid were 7 times higher than pinonic acid, similar to an earlier report
(Fu et al., 2009). The vapour pressure of pinic acid is about 2 orders of magnitude lower than
pinonic acid (Bhat and Fraser, 2007). Thus, pinic acid can saturate and readily nucleate, being
expected to have a higher fraction in the aerosol phase than pinonic acid.
Two novel monoterpene SOA tracers, 3-hydroxyglutaric acid (3-HGA) and 3-methyl-1,2,3-
butanetricarboxylic acid (MBTCA), were also detected in these marine aerosols. Both of them
are reported to be higher-generation products of α-pinene photooxidation (Kourtchev et al.,
2009; Szmigielski et al., 2007). The concentration ranges of 3-HGA in marine aerosols over
ECS were lower than those of aerosols in Mt. Tai, central east China (Fu et al., 2010), but
comparable to other studies about marine aerosols (Fu et al., 2011). The average abundance of
atmospheric MBTCA for all samples were 5.6 ng m$^{-3}$ with mean concentrations of 6.8 ng m$^{-3}$
and 4.5 ng m$^{-3}$ during the day and night, respectively, more abundant than 3-HGA (2.2 ng m$^{-3}$
). Interestingly, these mean values of MBTCA were comparable to those of mountain
aerosols (Fu et al., 2010), but still higher than the marine aerosols collected on the remote
seas (Fu et al., 2011). Since the ECS is adjacent to Mainland China, the atmospheric boundary
layer in these regions would inevitably be affected by continent-derived air masses, as
supported by the spatial pattern of individual SOA tracers (Fig. 4 and Fig. S1), in particular of
the peaks occurring in the coastal areas. Furthermore, in contrast to isoprene SOA tracers,
monoterpene SOA tracers commonly exhibit higher average daytime concentrations (Table 1
and Fig. 5), but T-test showed no significant difference ($p > 0.05$) between daytime and
nighttime concentrations for both total and individual monoterpene SOA tracers. The
concentration ratio of 3-HGA plus MBTCA to pinic acid ((3HGA+MBTCA)/pinic) showed
higher values in the terrestrially influenced aerosols (Fig. 6c). Besides, this mass
concentration ratios were basically higher during the daytime (mean 2.0) than nighttime (1.9),
indicating that aged aerosols are more abundant in the daytime.

### 3.2.3 Sesquiterpene SOA tracers

The analysis of sesquiterpenes is a great challenge due to their high reactivity and relatively
low vapour pressure. However, the aging of β-caryophyllene derived SOA has proved its
significant effects on all CCN-relevant properties (Asa-Awuku et al., 2009). β-caryophyllene
is one of the most abundant species among sesquiterpenes originating from plants (Duhl et al.,
2008). β-caryophyllinic acid is an ozonolysis or photooxidation product of β-caryophyllene
(Jaoui et al., 2007). Moreover, sesquiterpenes accumulated in leaves and woods can be
emitted during biomass combustion (Ciccioli et al., 2014).
The atmospheric levels of β-caryophyllinic acid were 0.16–17.2 ng m$^{-3}$ (mean 2.9 ng m$^{-3}$),
with 0.17–17.2 ng m$^{-3}$ (3.5 ng m$^{-3}$) during daytime and 0.16–9.6 ng m$^{-3}$ (2.3 ng m$^{-3}$) during
nighttime, respectively (Table 1). As expected, β-caryophyllinic acid correlated well with
levoglucosan ($p < 0.001$, $r = 0.61$, N = 51) in agreement with earlier report (Ding et al., 2016),
indicating substantial contribution of terrestrial biomass burning to sesquiterpenes' loading in
the marine atmosphere. On the other hand, biomass-burning processes (e.g. crop straw
combustion and forest fires) can raise ambient and/or leaf temperature to some extent,
consequently probably enhancing the emission of sesquiterpenes from trees and plants.
Many researches have proved that sesquiterpene emissions increase significantly with
increasing ambient temperatures (Tarvainen et al., 2005; Jaoui et al., 2007). Ambient
temperature seems to be the dominant factor controlling temporal variation in sesquiterpene
emission although other factors contribute (Duhl et al., 2008). An early study reported that
emission rates of sesquiterpene were 1.2–3 times higher in the daytime as well (Duhl et al.,
2008). All the reports described above perfectly interpret much higher concentrations of β-
caryophyllinic acid during daytime in our study. The abundance of β-caryophyllinic acid in
this study are higher than those reported in the remote marine aerosols during a round-the-
world cruise, but comparable to the maximum concentration of 2.5 ng m$^{-3}$ observed at
California coast (Fu et al., 2011); however, the concentrations of β-caryophyllinic acid were
much lower than those of Mt. Tai aerosols, central east China (average 12 ng m$^{-3}$ for both
daytime and nighttime aerosols) (Fu et al., 2010). The spatial distributions of β-caryophyllinic
acid also presented higher levels in coastal regions than other sampling sites (Fig. 4h), again
proving contribution of terrestrial aerosols.

### 3.3    Contributions of biogenic primary and secondary sources

To assess the relative abundances of organic aerosols from primary and secondary emission
sources, tracer-based methods are employed to evaluate their contributions to the marine
ambient OC. For example, mannitol and arabitol determined in marine aerosols were used to
calculate the contributions of fungal spores to OC (Bauer et al., 2008); the average mass
percent ratios of levoglucosan to OC (8.14%) are used to investigate the biomass burning
derived OC (Fu et al., 2014). Besides, biogenic SOA tracers detected in the present study are
utilized to evaluate the SOC formation resulting from the oxidation of isoprene, α-pinene and
β-caryophyllene through a tracer-based method reported by Kleindienst et al. (2007). This
method used the laboratory-derived mass fractions ($f_{soc}$) of marker species generated from
known precursors into SOC concentrations. Specifically, the $f_{soc}$ values used in our study is
0.155±0.039 for isoprene, 0.231±0.111 for α-pinene and 0.023±0.005 for β-caryophyllene,
respectively (Kleindienst et al., 2007). Through dividing the sum of tracer compounds
measured in these marine samples by $f_{soc}$, an estimate of the contribution of each SOA to the
total marine SOC concentration was determined and the results were presented in Table 2.
Biomass burning derived OC in the whole samples is in the range of 1.1–790 ngC m$^{-3}$ (89.6
ngC m$^{-3}$), with higher levels in the nighttime (1.3–790 ngC m$^{-3}$, 103 ngC m$^{-3}$) compared to
those (1.1–506 ngC m$^{-3}$, 75.5 ngC m$^{-3}$) in the daytime. Higher concentrations of biomass-
burning OC near the Asian continent than those over the remote oceans (Fig. 7) suggest that
continental biomass-burning tracers were possibly removed by dry and/or wet deposition of

airborne particles, photodegradation by free radicals in the atmosphere or other atmospheric dilution mechanism during long-range transport to the western North Pacific. Fungal-spore-derived OC for all samples ranged from 1.2–1840 ngC m$^{-3}$ (180 ngC m$^{-3}$), accounting for 0.03–19.8% (3.1%) of OC, higher than those of biomass-burning-derived OC (Table 2). The nighttime fungal-spore-derived OC (1.2–1840 ngC m$^{-3}$, 203 ngC m$^{-3}$) is higher than those of daytime ones (1.3–911 ngC m$^{-3}$, 157 ngC m$^{-3}$), possibly associated with intensified activities of yeasts and fungi during nighttime (Fu et al., 2012). As we all known, evenings tend to have higher moist content, which exerts a stronger influence on microbial activity than temperature (Liang et al., 2003). Similar higher levels of fungal-spore-derived OC were also observed in the aerosols collected in coastline waters, probably due to more intensified microbial activities in the coastal regions as discussed previously. β-caryophyllene SOC (6.9–747 ngC m$^{-3}$, 126 ngC m$^{-3}$, with 7.4–747 ngC m$^{-3}$, mean 153 ngC m$^{-3}$ during the day and 6.9–416 ngC m$^{-3}$, mean 100 ngC m$^{-3}$ at night) is found to be dominant contributor among the measured biogenic SOCs over the ECS (the average concentrations were 39.8 ngC m$^{-3}$ and 50.2 ngC m$^{-3}$ for isoprene SOC and α/β-pinene SOC, respectively). Moreover, β-caryophyllene SOC account for 0.36–5.3% (2.4%) of OC, about 2–3 times as high as those of isoprene (0.13–3.8%, 0.83%) and monoterpene SOC (0.08–3.5%, 0.98%). Therefore, an emission inventory for sesquiterpenes in marine aerosols over the ECS will be helpful for further understanding formation of biogenic SOA in this region. Zhu et al. (2016) also reported that the sesquiterpene-derived SOC was more abundant than isoprene- and monoterpene-derived SOC for the aerosols collected in Okinawa, Japan.

The higher levels of sesquiterpene-SOC than monoterpene- and isoprene-SOC may be due to the differences in the gas/particle partitioning of oxidation products from different VOCs, given that longer chain sesquiterpenes ($C_{15}H_{24}$) have more carbon atoms than monoterpenes ($C_{10}H_{16}$) and isoprene ($C_5H_8$), decreasing the vapour pressures of their oxidation products (Fu et al., 2016). The levels of SOCs stemmed from isoprene, monoterpenes and sesquiterpene in the marine aerosols over the ECS were much lower than those of PM$_{2.5}$ samples in Hong Kong, China during summer (Hu et al., 2008) and those observed in Mt. Tai aerosols (Fu et al., 2010). The sum of biogenic SOCs over the ECS is 11.3–1060 ngC m$^{-3}$ (216 ngC m$^{-3}$), much lower than that of Mt. Fuji aerosols (227–1120 ngC m$^{-3}$, 542 ngC m$^{-3}$) (Fu et al., 2014) and those in the Mt. Tai aerosols (420–3100 ngC m$^{-3}$) (Fu et al., 2010), but higher than those observed in marine aerosols collected during a round-the-world cruise covering most remote oceans (Fu et al., 2011) and those reported in the Arctic aerosols (average 14.6 ngC m$^{-3}$) (Fu

et al., 2009). Such difference between terrestrial aerosols and marine aerosols highlights the
outflow of continental aerosols again. In terms of spatial distributions, biogenic SOCs
calculated from these compounds showed higher loadings in the locations close to the
coastline or significantly influenced by terrestrial air (Fig. 8a-b), also validating strong
influence of continental origin. Generally, the total concentrations of biogenic SOC (216 ngC
$m^{-3}$) are higher than those of biomass-burning-derived OC (89.6 ngC $m^{-3}$) and fungal-spore-
derived OC (180 ngC $m^{-3}$) (Table 2), presenting greater contribution from biogenic SOA to
the marine aerosols, in agreement with the report by Fu et al. (2014).
In order to compare the relative contribution of marine and continental sources to total OC in
the oceanic atmosphere, the whole sampling area was divided into five regions from north to
south according to the spatial distribution of aerosol samples, i.e. northern waters of ECS,
nearby waters in the ECS, seas adjacent to Fujian and Zhejiang provinces, eastern waters of
Taiwan. Five-day HYSPLIT back trajectory analysis showed that the atmosphere over waters
north of ECS were mainly controlled by marine air masses, while aerosol samples achieved
on June 12 showed great influence from South Korea and North China as well (Fig. S2).
Aerosol samples in waters of ECS were basically under the control of marine air masses,
except for the samples collected on May 20, which were also affected by terrestrial air from
the Asian mainland (Fig. S3). In general, aerosols collected off the eastern coast of Taiwan
Island were affected by air masses from the remote sea, but some aerosols collected on May
22–23 were also influenced by air masses coming from Southeast Asia (Fig. S4). In contrast,
air masses from Asian mainland had substantial impacts on the samples collected near Fujian
and Zhejiang province (Fig. S5-S6). On the whole, the aerosols strongly affected by terrestrial
sources (e.g. Asian mainland, Southeast Asia and South Korea) tend to own higher levels of
sugars and SOA tracers (Fig. 2 and Fig. 4).
The contributions of biomass-burning-derived OC, fungal-spore-derived OC, and biogenic
SOCs to OC (%) in these five sampling regions were presented in Fig. 9. Generally, the
average contributions of biogenic SOCs, biomass-burning OC and fungal-spore OC to OC
near Zhejiang and Fujian waters were higher than the other sampling areas, especially for the
fungal-spores-OC (7.5±6.8 for Zhejiang waters, 4.0±2.8 for Fujian waters, 2.6±3.2 for the
northern waters of ECS, 1.9±1.7 for eastern waters of Taiwan, and 1.5±1.4 for ECS,
respectively) and biomass-burning-OC (2.8±2.6 for Zhejiang waters, 2.7±1.8 for Fujian
waters, 1.3±0.92 for the northern waters of ECS, 0.52±0.36 for eastern waters of Taiwan, and
0.96±1.7 for ECS, respectively). Such spatial variations were closely associated with different
contributions of land and marine sources to the oceanic atmosphere.
Figures S2-S6 display a strong influence from land air masses in waters around Zhejiang and
Fujian provinces, while the atmosphere over northern waters of ECS, ECS and eastern waters
of Taiwan Island basically came under the influence of relatively clean marine air.
Interestingly, the average percentage of isoprene SOC in the eastern waters of Taiwan was
slightly larger compared to other regions. In light of back trajectory analysis, aerosols in this
region were mainly affected by terrestrial photosynthetic vegetation (e.g. trees and plants) in
Southeast Asia and/or marine biota (e.g. phytoplankton, seaweed and bacteria) (Fig. S4).
Sesquiterpene-derived SOC was found to be the most abundant SOC species in all five areas
(2.9±0.87 for Zhejiang waters, 3.1±1.7 for Fujian waters, 2.1±0.80 for the northern waters of
ECS, 2.2±1.2 for eastern waters of Taiwan, and 2.2±1.3 for ECS, respectively) in comparison
with other SOCs derived from isoprene and monoterpene. On the other hand, the nighttime
contributions of biomass-burning OC, fungal-spores OC and biogenic SOCs to OC were
commonly greater than the daytime ones in all five regions (Fig. S9). For instance, the
contributions of biomass-burning-OC and fungal-spores-OC to OC during nighttime in the
seas near Zhejiang (3.9±3.5 and 9.3±8.0, respectively) are significantly greater than those in
the daytime (1.9±1.2 and 6.0±6.3, respectively). Such enhanced contributions in the evening
were likely to be in connection with intensified emissions, decreased height of PBL and land-
sea breeze circulations at night. The downward movement of PBL does not facilitate
dispersion of pollutants and lead to increases in aerosol concentration in the lower PBL (Li et
al., 2017). The prevailing land breeze in the nighttime in coastal areas could bring plentiful
terrestrial particles to the clean marine atmosphere. Such difference between daytime and
nighttime contributions illustrates that land-sea breeze circulation and PBL can be another
important factor influencing organics in marine aerosols.

## 3.4  Source apportionment by PMF

### 3.4.1  Analysis of source profiles

After testing runs with different number of factors (5–9), eight factors were chosen on basis of
the minimum value of Q (goodness of fit parameters) and probable source profile expected
from the study region.
Figure 10 illustrates the first source had high loadings of $Na^+$ (90.8%), suggesting a
contribution from sea salt. This profile also contained a significant amount of $SO_4^{2-}$, which
can react with sea-salt particles and release HCl gas, leaving lower $Cl^-/Na^+$ ratio (0.1) than
that of sea water (1.8) (Boreddy et al., 2014).
The second source shows high loadings of levoglucosan (66.3%), arabitol (55.0%), mannitol
(51.1%) and trehalose (50.4%), representing mixed sources of biomass burning and fungal
spores (Simoneit et al., 1999; Bauer et al., 2008; Medeiros et al., 2006a). Yang et al. (2012)
found an enhanced abundance of fungal tracers on account of biomass burning activities,
during which large numbers of fungi could be dispersed into the surrounding atmosphere or
be carried upward with the warm plume to other fields via long-distance atmospheric
transport.
The third source can be interpreted as crustal dust because of high loadings of $Mg^{2+}$ (66.1%)
and $Ca^{2+}$ (64.2%), characteristic elements of soil/crustal dust (Xu et al., 2016). This source
may include airborne road dust, construction dust and windblown soil particles, which are
derived primarily from terrestrial source.
The fourth factor is characterized by dominance of $NH_4^+$ (58.4%) and $SO_4^{2-}$ (49.7%), which
can be classified as secondary ammonium sulfate. The molar ratio of $NH_4^+$ and $SO_4^{2-}$ was 2.9
in this profile, suggesting that $(NH_4)_2SO_4$ was the dominated sulfate form in the marine
atmosphere over the ECS. Due to its long lifetime in the atmosphere, terrestrial $SO_4^{2-}$ could
be transported long distance to coastal areas and even to the remote sea (Itahashi et al., 2017),
affecting chemical composition of oceanic atmosphere.
The fifth factor exhibits high loadings of di-isobutyl (DiBP) and di-n-butyl (DnBP), dominant
species of phthalates in the marine aerosols over the ECS, which is assigned to plastic
materials' emission likely from industry, agriculture and domestic application in coastal
regions.
The sixth factor has high loadings from sucrose (85.3%) followed by fructose (50.9%),
implies a significant emission from airborne pollen grains to the marine atmosphere over the
ECS during late spring to early summer (Fu et al., 2012).
The seventh profile presents high loading of Cl, likely to be associated with coal combustion,
which provides significant releases of chlorine (McCulloch et al., 1999; Sun et al., 2013).
Another confirmation is high concentration of $SO_4^{2-}$ from this profile, since coal consumption
can produce mass sulfate as well.
The eighth factor illustrates high loadings of $NO_3^-$ (67.5%) and biogenic SOA tracers,
especially the monoterpene SOA tracers, i.e. PA (67.5%) and MBTCA (65.2%). This factor
could be attributed to the photochemical oxidation products stemmed from emission of
vehicle and biogenic VOCs. Formation of secondary nitrate depends on $NO_x$, which is mainly
produced from power plants and mobile sources (Heo et al., 2009; Kim et al., 2006).
Anthropogenic $NO_x$ could also enhance biogenic SOA formation via nitrate radical oxidation
of monoterpenes (Xu et al., 2015). The positive correlations between $NO_x$ and isoprene-
derived SOA tracers as discussed before in our study also suggest $NO_3^-$ and BSOA may share
common formation and/or transport pathways. Previous studies have reported that nitrogen-
containing species act a pivotal part in the formation and fate of SOA through varying radical
and oxidant regimes and particle properties, such as volatility and hygroscopicity (Chen et al.,
2017), agreeing well with our results.
Overall, the eight sources based on PMF were sea salt, biomass burning and fungal spores,
crustal dust, secondary sulfate ammonia, plastic emission, pollen grains, coal combustion, and
secondary nitrate and BSOA, which contributed to the TSP over the ECS of 16.9%, 3.5%,
8.0%, 28.5%, 6.3%, 2.0%, 14.3% and 20.4%, respectively. The results of PMF present that
secondary origin and marine natural emissions could be the main sources for the aerosols over
the ECS. Figure 11 shows the contributions of different sources to OC in marine aerosols. On
the whole, secondary nitrate and BSOA (25.5%), and biomass burning and fungal spores
(19.5 %) contributed more to OC than other sources during the whole sampling periods,
elucidating the significant influence of biogenic primary and secondary sources on marine
organic aerosols.

## 3.4.2 Temporal and spatial variation in sources

Figure 12 shows the temporal and spatial variation in each profile during the day and the night.
The contribution of each source changes over time and varies with distance from the continent.
In general, higher levels contributed by biomass burning, fungal spores, crustal dust, pollen
grains, coal combustion, secondary nitrate and BSOA, were basically observed in coastal
aerosols and/or terrestrially influenced aerosols, suggesting strong influence of continental air
from East Asia and Southeast Asia in light of the back trajectories and wind directions during
the sampling periods (Fig. S2-S6 and S10). However, the contributions from sea-salt particles,
indicative of oceanic emission (organic components can be emitted from the ocean surface
together with sea-salt particles via sea spray or bubble bursting), tend to be higher in the

aerosols mainly affected by marine air masses. Our study demonstrates that primary and secondary OM of terrestrial origin play an important role in the marine aerosol chemistry over the western North Pacific through long-range atmospheric transport in addition to natural emission of ocean.

# 4   Conclusions

In summary, atmospheric concentration, spatial distribution and source apportionment of sugars and biogenic SOA tracers were studied for the coastal and remote marine aerosols. Higher concentrations of sugars and BSOA tracers were observed in the atmosphere around coastal waters and/or in the terrestrially influenced regions in comparison with the remote oceans, suggesting that continent origin contributed a lot to the abundance of sugars and BSOA tracers in the marine atmosphere. Glucose was the dominant sugar species, followed by mannitol among the total identified sugar compounds. Biogenic SOC were characterized by a predominance of β-caryophyllene oxidation products in comparison with isoprene and α/β-pinene tracers. The contributions of biomass-burning-derived OC, fungal-spore-derived OC, and biogenic SOC to OC (%) were greater in the marine aerosols affected by land air masses. The results of PMF illustrate that secondary nitrate, BSOA, biomass burning, and fungal spores could be the major contributors to OC in marine aerosols over the ECS. Our study demonstrates that both primary and secondary organic aerosols of terrestrial origin have great influences on the marine aerosol chemistry over the western North Pacific through long-range atmospheric transport.

***Data availability***. The data for this paper are available upon request from the corresponding author (fupingqing@tju.edu.cn).

***Competing interests***. The authors declare that they have no conflict of interest.

**Acknowledgements**

This study was supported by the National Natural Science Foundation of China (Grant Nos.
41625014, 41475117, 41571130024 and 91543205). We are grateful to the crew of the marine
cruise supported by the National Natural Science Foundation of China.

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

**Table 1.** Concentrations (ng m$^{-3}$) of saccharides and biogenic SOA tracers measured in the
marine aerosols collected over the East China Sea during May to June 2014.

| Compounds | Whole period (n = 51) | | Daytime (n = 25) | | Nighttime (n = 26) | |
|---|---|---|---|---|---|---|
| | Mean | Range | Mean | Range | Mean | Range |
| *Anhydrosugars* | | | | | | |
| Galactosan | 0.59 | 0.02–3.9 | 0.56 | 0.04–3.6 | 0.62 | 0.02–3.9 |
| Mannosan | 1.4 | 0.06–10.2 | 1.2 | 0.08–5.2 | 1.6 | 0.06–10.2 |
| Levoglucosan | 7.3 | 0.09–64.3 | 6.1 | 0.09–41.2 | 8.4 | 0.10–64.3 |
| *Sugar Alcohols* | | | | | | |
| Erythritol | 1.3 | n.d.$^a$–16.9 | 0.87 | 0.02–4.60 | 1.6 | n.d.–16.9 |
| Arabitol | 5.1 | 0.02–51.0 | 4.5 | 0.06–26.7 | 5.7 | 0.02–51.0 |
| Mannitol | 16.3 | 0.03–169 | 14.1 | 0.08–81.3 | 18.4 | 0.03–169 |
| Inositol | 0.57 | 0.01–7.7 | 0.47 | 0.01–2.8 | 0.67 | 0.01–7.7 |
| *Sugars* | | | | | | |
| Fructose | 9.1 | 0.09–106 | 8.1 | 0.09–56.5 | 10.2 | 0.32–106 |
| Glucose | 18.8 | 0.31–209 | 17.8 | 0.31–106 | 19.7 | 0.47–209 |
| Sucrose | 11.7 | 0.09–216 | 9.7 | 0.14–136 | 13.5 | 0.09–216 |
| Trehalose | 9.4 | 0.08–96.0 | 9.1 | 0.08–60.7 | 9.7 | 0.10–96.0 |
| Subtotal | 81.5 | 1.8–950 | 72.6 | 1.0–525 | 90.1 | 1.8–950 |
| *Isoprene SOA tracers* | | | | | | |
| 2-methylthreitol | 1.5 | 0.01–12.7 | 1.2 | 0.04–5.2 | 1.9 | 0.01-12.7 |
| 2-methylerythritol | 3.2 | 0.02–29.1 | 2.5 | 0.08–11.8 | 3.9 | 0.02–29.1 |
| Sum of 2-methyltetrols | 4.8 | 0.03–41.9 | 3.8 | 0.11–17.0 | 5.7 | 0.03–41.9 |
| 2-Methylglyceric acid (2-MGA) | 1.4 | 0.09–8.3 | 1.2 | 0.20–3.6 | 1.6 | 0.09–8.33 |
| $C_5$-Alkene triols $^b$ | 2.2 | 0.03–14.6 | 2.1 | 0.06–14.6 | 2.3 | 0.03–13.8 |
| Subtotal | 8.4 | 0.15–64.0 | 7.1 | 0.42–35.2 | 9.6 | 0.15–64.0 |
| *α/β-pinene (monoterpene) SOA tracers* | | | | | | |
| 3-Hydroxyglutaric acid (3-HGA) | 2.2 | 0.03–14.1 | 2.3 | 0.08–14.1 | 2.1 | 0.03–7.7 |
| Pinonic acid (PNA) | 0.47 | 0.02–1.6 | 0.49 | 0.06–1.6 | 0.46 | 0.02–1.6 |
| Pinic acid (PA) | 3.4 | 0.16–14.9 | 3.4 | 0.28–14.9 | 3.3 | 0.16–13.6 |
| MBTCA | 5.6 | n.d.–56.9 | 6.8 | n.d.–56.9 | 4.5 | n.d.–24.5 |
| Subtotal | 11.6 | 0.26–87.2 | 13.0 | 0.45–87.2 | 10.3 | 0.26–43.7 |
| *β-caryophyllene (sesquiterpene) SOA tracers* | | | | | | |
| β-Caryophyllinic acid | 2.9 | 0.16–17.2 | 3.5 | 0.17–17.2 | 2.3 | 0.16–9.6 |
| Total measured tracers | 22.9 | 1.1–135 | 23.6 | 1.4–135 | 22.2 | 1.1–115 |

$^a$ n.d. denotes not detected. $^b$ $C_5$-alkene triols: cis-2-methyl-1,3,4-trihydroxy-1-butene,
3-methyl-2,3,4-trihydroxy-1-butene and trans-2-methyl-1,3,4-trihydroxy-1-butene.


**Table 2.** Concentrations of organic carbon (OC) (ngC m$^{-3}$) from biogenic primary emission (biomass-burning OC and fungal-spore OC) and biogenic SOC and their contributions in aerosol OC (%) in marine aerosols over the East China Sea.

| Component | Total | | | Daytime | | | Nighttime | | |
|---|---|---|---|---|---|---|---|---|---|
| | Range | Mean | std | Range | Mean | std | Range | Mean | std |
| Concentration (ngC m$^{-3}$) | | | | | | | | | |
| Aerosol OC | 424–14100 | 4260 | 3480 | 1030–14100 | 4940 | 3800 | 424–9830 | 3600 | 3080 |
| Biomass burning OC | 1.1–790 | 89.6 | 156 | 1.1–506 | 75.5 | 117 | 1.3–790 | 103 | 187 |
| Fungal spore OC | 1.2–1840 | 180 | 334 | 1.3–911 | 157 | 249 | 1.2–1840 | 203 | 403 |
| Isoprene SOC [a] | 0.78–324 | 39.8 | 55.4 | 2.3–133 | 32.1 | 37.6 | 0.78–324 | 47.2 | 68.3 |
| Monoterpene SOC | 1.1–377 | 50.2 | 72.0 | 1.9–377 | 56.1 | 84.3 | 1.1–189 | 44.6 | 58.9 |
| Sesquiterpene SOC | 6.9–747 | 126 | 153 | 7.4–747 | 153 | 183 | 6.9–416 | 100 | 114 |
| Sum of biogenic SOC | 11.3–1060 | 216 | 250 | 13.7–1060 | 241 | 279 | 11.3–824 | 192 | 222 |
| Subtotal | 16.1–3460 | 486 | 688 | 29.8–2480 | 473 | 600 | 16.1–3460 | 498 | 776 |
| Percentage in aerosol OC (%) | | | | | | | | | |
| Biomass burning OC | 0.05–8.5 | 1.5 | 1.8 | 0.05–4.4 | 1.1 | 1.2 | 0.13–8.5 | 1.8 | 2.3 |
| Fungal spore OC | 0.03–19.8 | 3.1 | 4.0 | 0.03–16.6 | 2.6 | 3.7 | 0.18–19.7 | 3.5 | 4.2 |
| Isoprene SOC | 0.13–3.8 | 0.83 | 0.87 | 0.13–2.0 | 0.60 | 0.54 | 0.14–3.8 | 1.0 | 1.1 |
| Monoterpene SOC | 0.08–3.5 | 0.98 | 0.90 | 0.10–3.3 | 1.00 | 0.95 | 0.08–3.5 | 0.96 | 0.87 |
| Sesquiterpene SOC | 0.36–5.3 | 2.4 | 1.2 | 0.36–5.3 | 2.5 | 1.4 | 0.87–5.1 | 2.4 | 1.1 |
| Sum of biogenic SOC | 0.67–9.3 | 4.2 | 2.2 | 0.67–9.3 | 4.1 | 2.2 | 1.8–8.8 | 4.4 | 2.2 |
| Subtotal | 1.5–37.0 | 8.7 | 6.9 | 1.5–27.4 | 7.8 | 6.3 | 2.8–37.0 | 9.7 | 7.4 |

[a] The total mass concentrations of SOC produced by isoprene (2-methylglyceric acid and 2-methyltetrols were used), $\alpha/\beta$-pinene, and $\beta$-caryophyllene were estimated using the tracer-based method by Kleindienst et al. (2007).

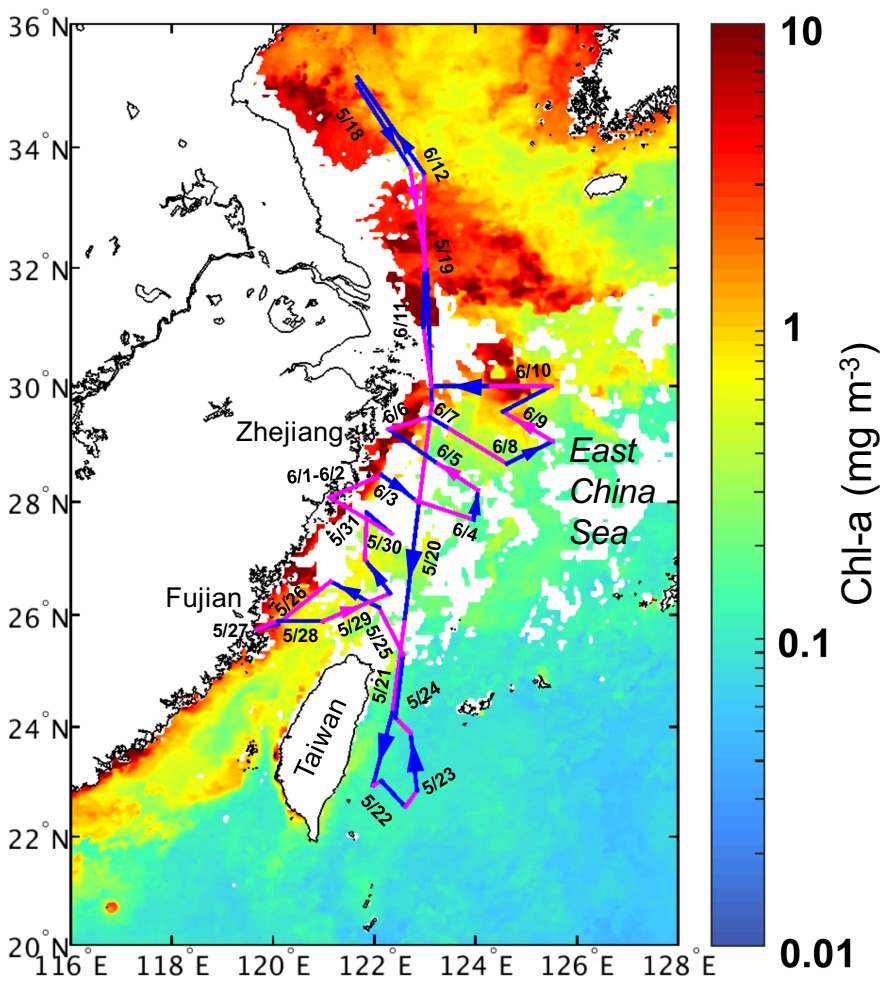



**Figure 1.** Cruise tracks of KEXUE-1 and spatial distribution of satellite-derived chlorophyll-a
concentrations (chl-a, mg m$^{-3}$) in surface seawater derived from MODIS L3 products during
the sampling period in the East China Sea. The purple and blue lines represent daytime and
nighttime aerosol samples, respectively.

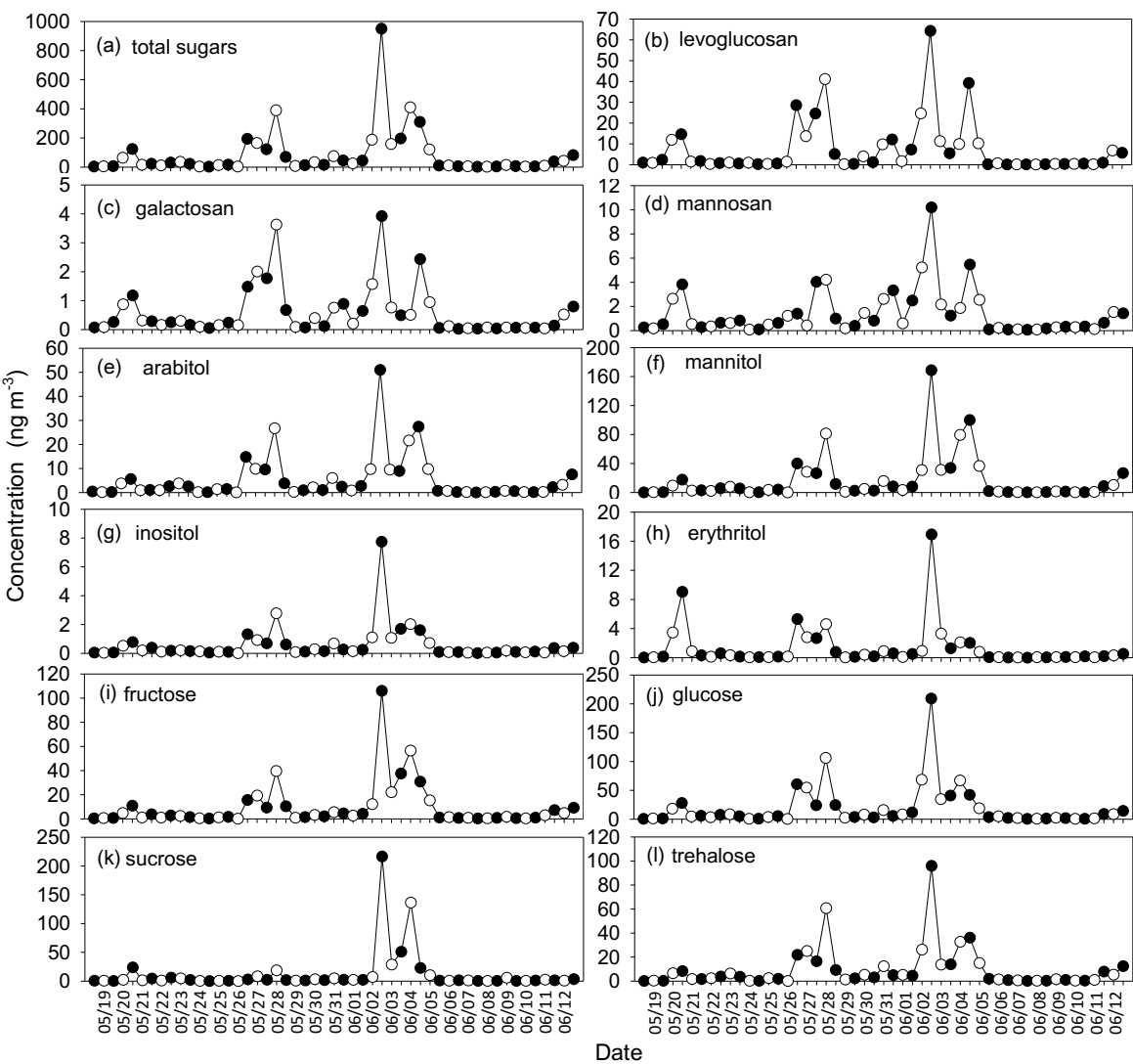


**Figure 2.** Temporal variations in sugar compounds in marine aerosols collected over the East
China Sea. The open and shaded circles represent daytime and nighttime values, respectively.

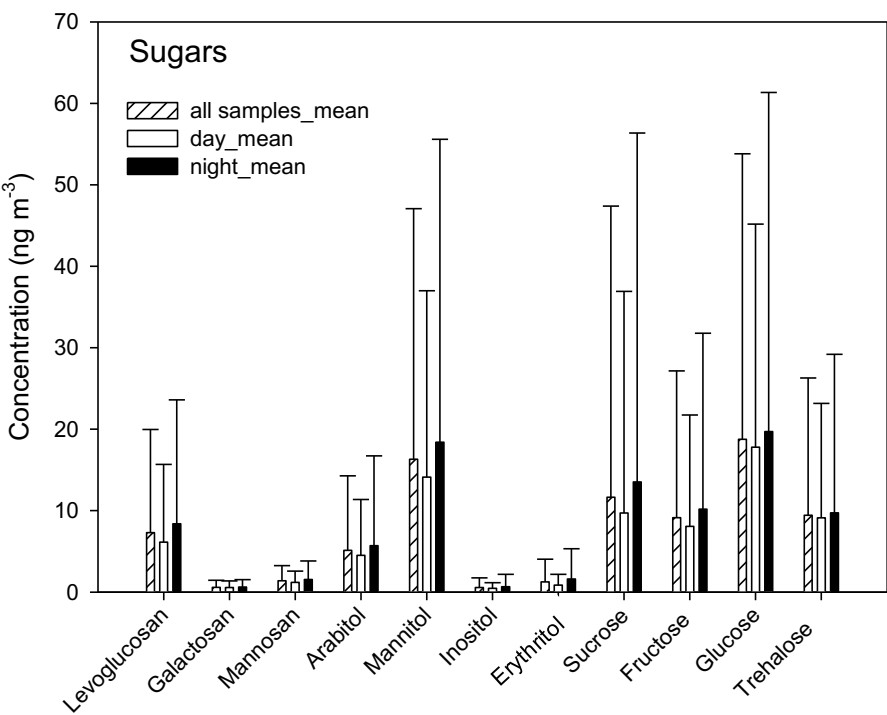



**Figure 3.** Average concentrations of sugars detected in marine aerosols over the East China
Sea.

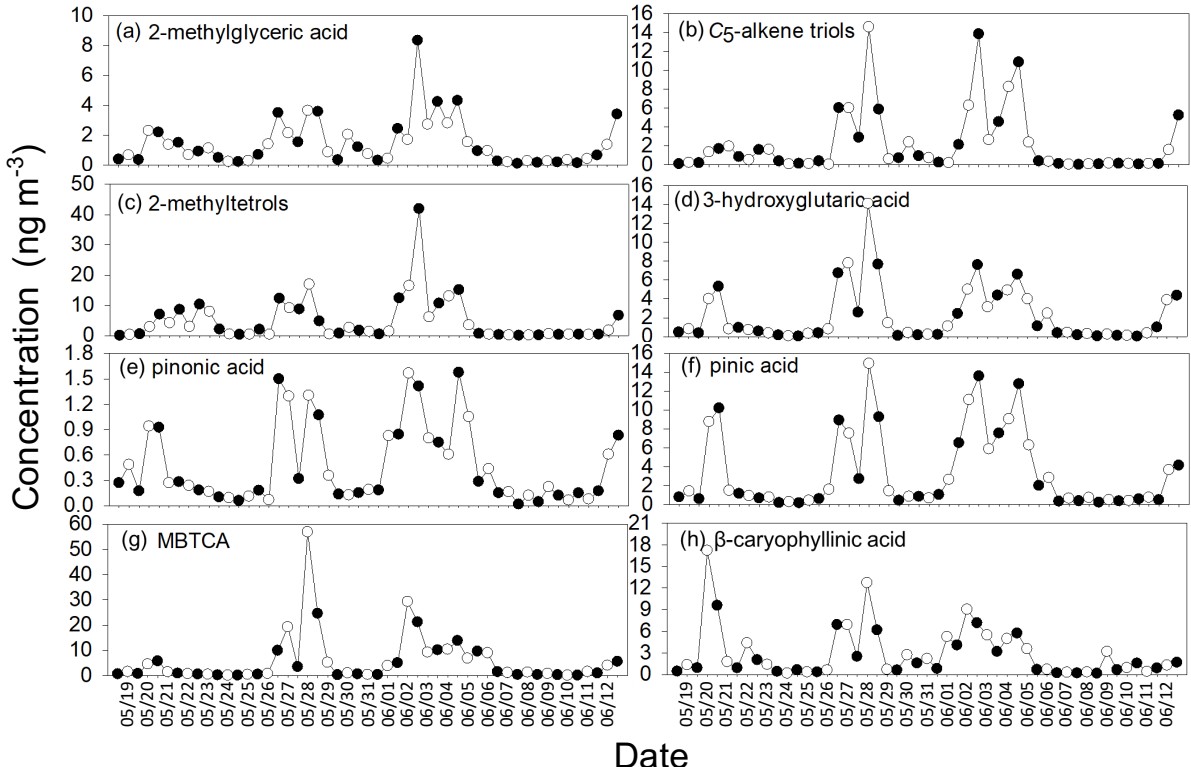


**Figure 4.** Temporal variations in biogenic SOA tracers detected in marine aerosols over the
East China Sea during May to June 2014. The open and shaded circles represent daytime and
nighttime values, respectively.


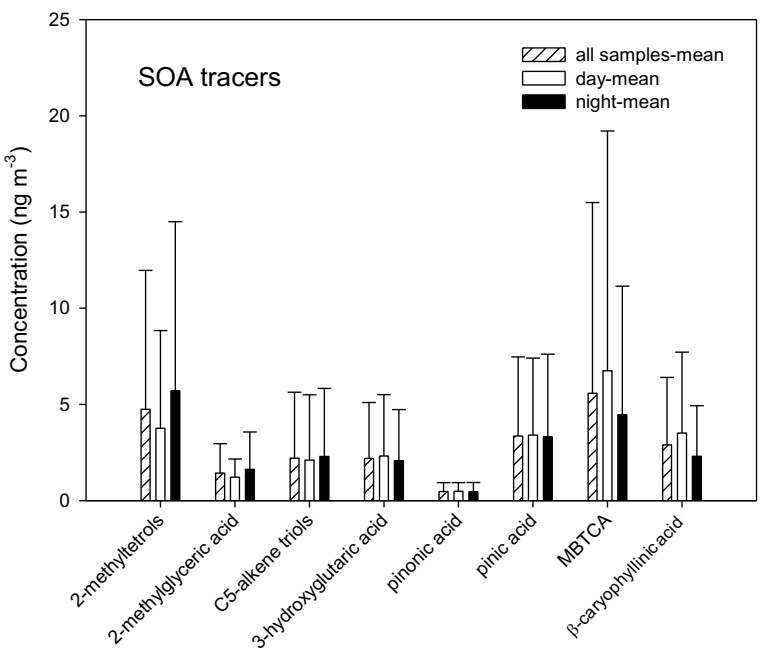


**Figure 5**. Average concentrations of SOA tracers detected in marine aerosols over the East
China Sea.


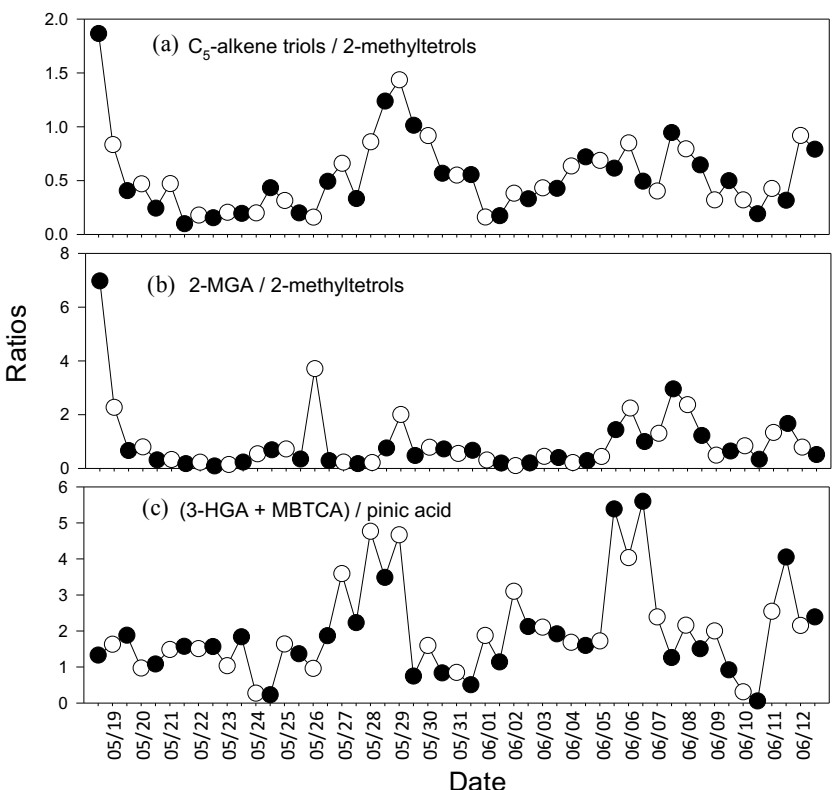


**Figure 6**. Temporal variations in the concentration ratios of isoprene and α/β-pinene oxidation
products in the marine aerosols over the East China Sea. The open and shaded circles
represent daytime and nighttime samples, respectively.

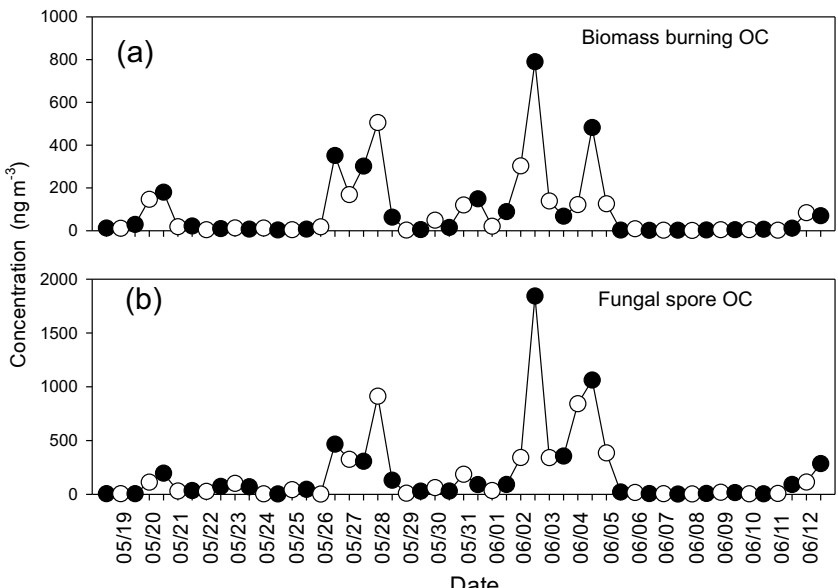



**Figure 7.** Temporal variations in (a) biomass-burning-derived OC, and (b) fungal-spore-
derived OC in the marine aerosols over the East China Sea. The open and shaded circles
represent daytime and nighttime samples, respectively.

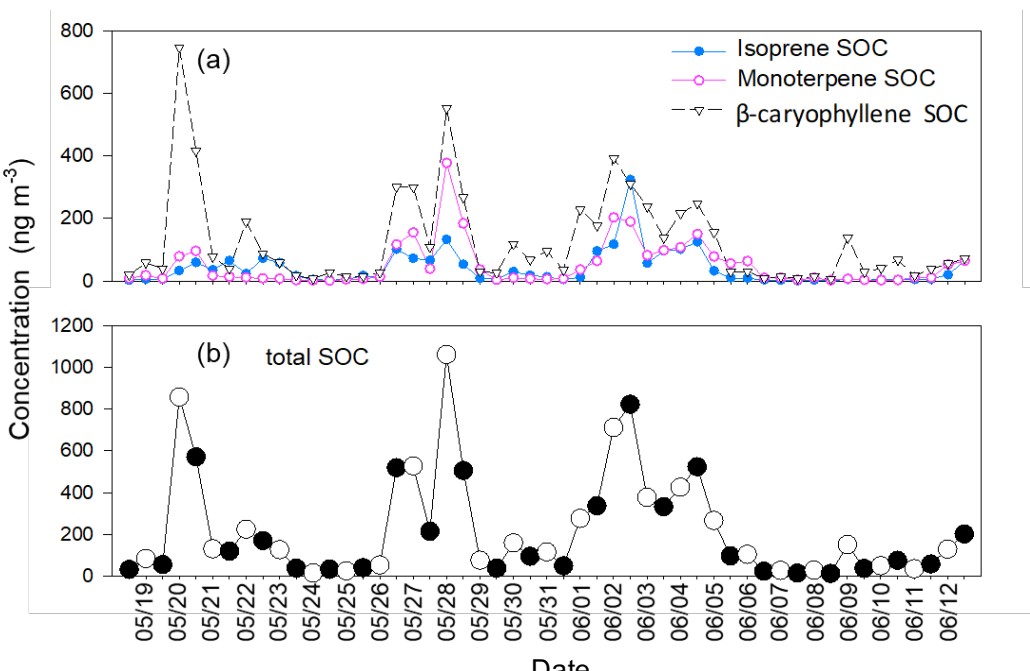



**Figure 8**. Temporal variations in (a) secondary organic carbon (SOC) derived from isoprene,
α/β-pinene and sesquiterpene and (b) the total SOC levels in the marine aerosols over the East
China Sea during May to June 2014. The open and shaded circles represent the daytime and
nighttime samples, respectively.

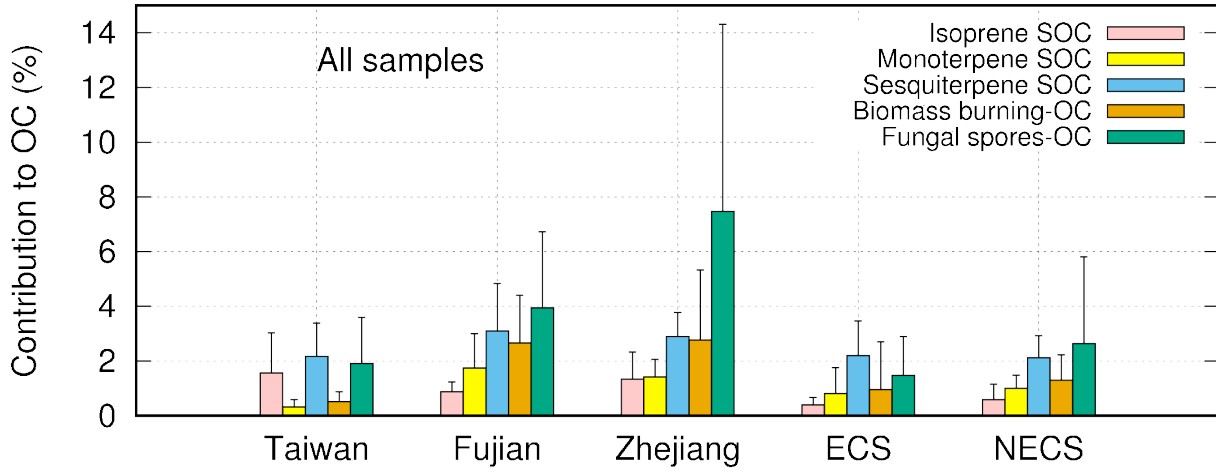



**Figure 9.** Contributions of OC (ngC m$^{-3}$) from biogenic primary emission (biomass-burning OC and fungal-spore OC) and biogenic SOC to OC (%) in different sampling regions. Taiwan, Fujian and Zhejiang refer to the waters around Taiwan, Fujian and Zhejiang; ECS represents East China Sea waters; NECS represents northern waters of ECS.

1186

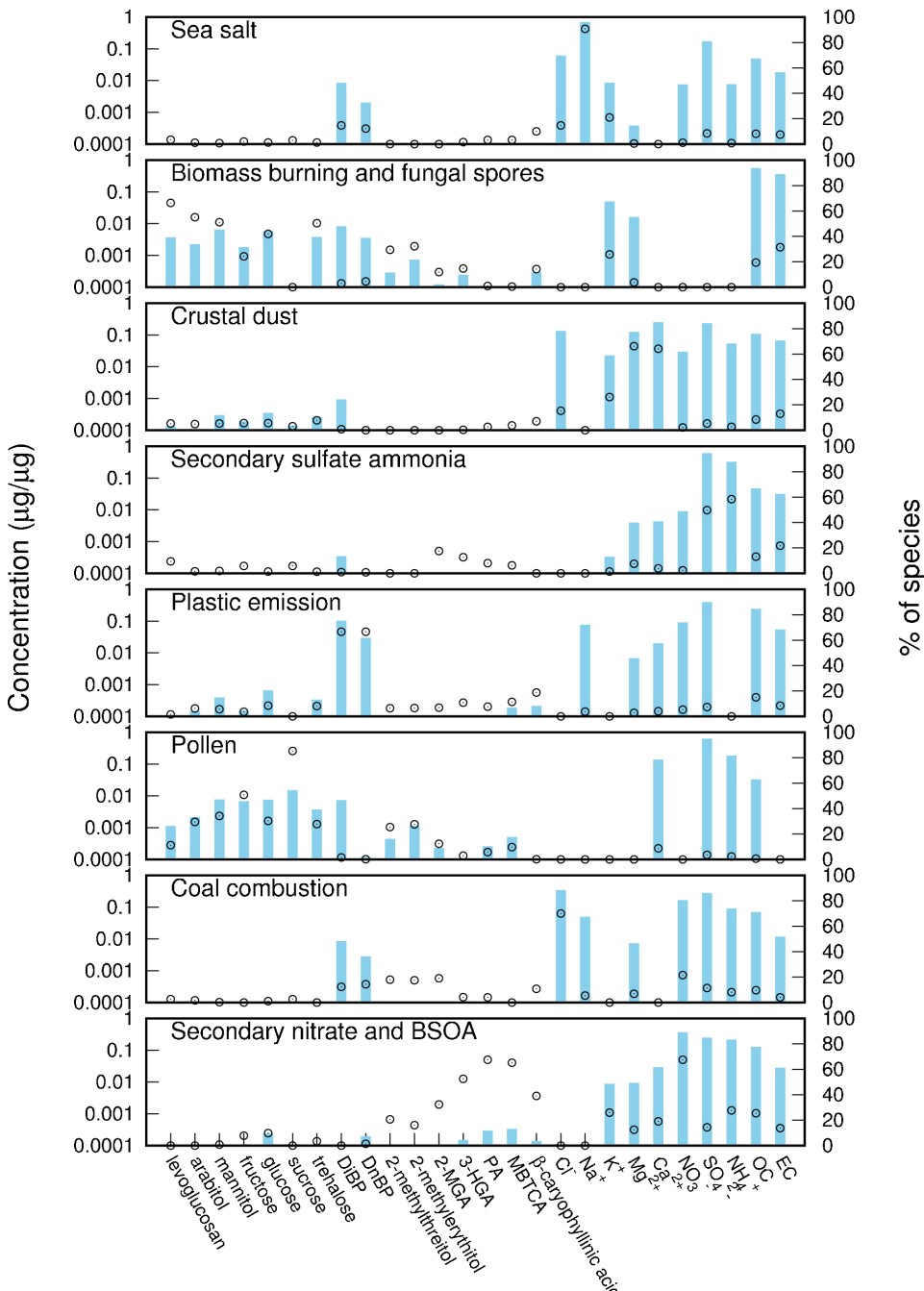

**Figure 10**. Source profiles identified by PMF. Blue bars represent the mass contribution with y-axis on the left, while black dots stand for the percentage of species to the sum with y-axis on the right.

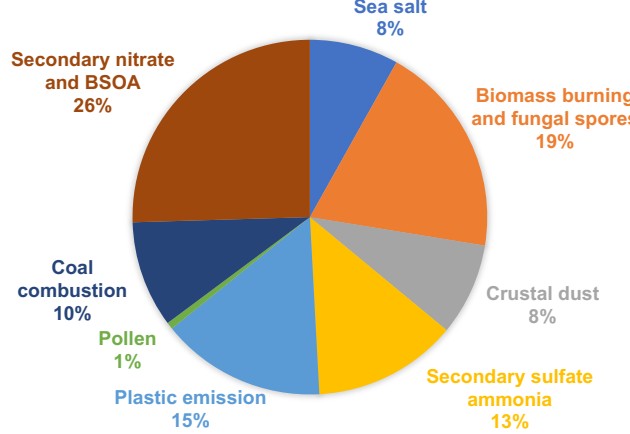



**Figure 11.** Contributions of different sources to organic carbon (OC) in marine aerosols over
the East China Sea.

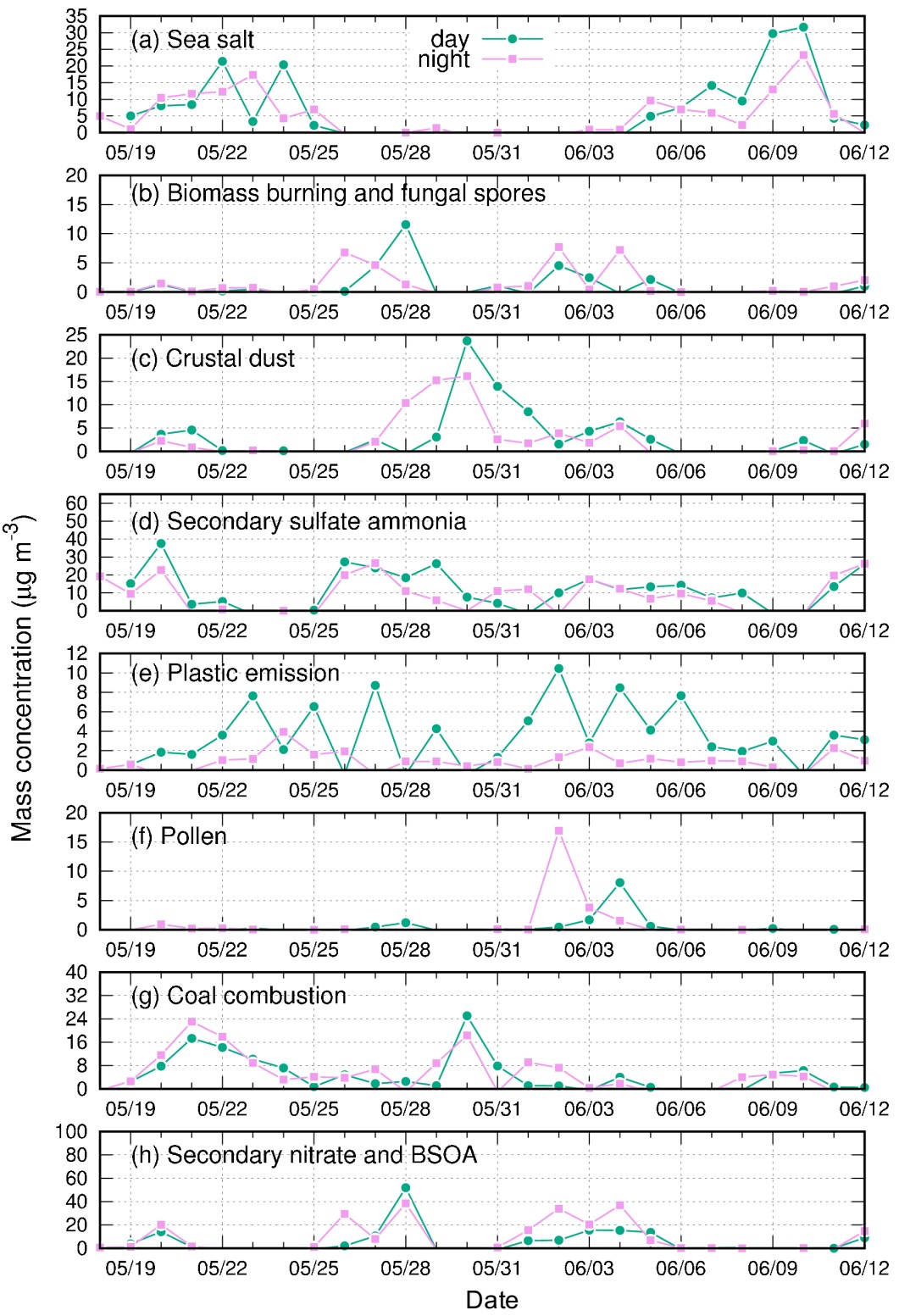



**Figure 12.** Temporal variation in sources contributed to marine aerosols ($\mu g\ m^{-3}$).