# Peer review of "Characterization of biogenic primary and secondary organic aerosols in the marine atmosphere over the East China Sea"

_Atmospheric Chemistry and Physics, 2018_

## Referee Comment (RC1) · Anonymous Referee #2 · 17 Jun 2018

This manuscript is one of the few studies to report polar primary and secondary organic compounds in marine aerosols. The authors took day and nighttime samples over the East China Sea for a period of about one month, and analyzed the samples for a series of biomass burning and biogenic secondary organic aerosol marker compounds. In this study, the authors highlighted that marine aerosols over the East China Sea, especially in the coastal regions were significantly affected by terrestrial emissions, and the emissions likely impact the air quality in the west pacific regions through transboundary flows of air masses. Overall the manuscript is straightforward and well written. I recommend the publication of this manuscript as is.

---

## Referee Comment (RC2) · Anonymous Referee #1 · 10 Jul 2018

This manuscripts reports on molecular composition, abundances and spatial distribution of sugars and biogenic secondary organic aerosols in the MABL of East China Sea (ECS). A total of 51 samples were collected in May-June 2014, day-time N=25 and night-time N=26. Authors state that (page 21, line 647) the contribution of each source changes over time and varies with distance from the continent. In this context, concentrations of the measured species are not representative on temporal and spatial scale for ECS and NW Pacific. Although the reported data set provides some novelty, there are number of issues with the discussion and interpretation presented by the authors.

[Figure]

For a general reader, authors should clearly state constituents referred as secondary organic aerosols (SOA) measured in this study. At times authors are addressing these constituents as "biogenic SOA tracers" and/or "biogenic SOA markers". This is confusing to a general reader.

Title: "Characterization of biogenic primary and secondary organic aerosols—-". What primary biogenic aerosols are being referred and measured in this study? Authors have mainly discussed biogenic secondary aerosols.

Abstract, lines 29-30: Authors state that "Biogenic SOA tracers and sugars exhibit higher levels in the samples affected by continental air masses, suggesting the terrestrial outflows of organic matter to the East China Sea". This is a redundant statement. If high concentrations of measured species are associated with continental air masses, then continental outflow of organic matter is much obvious (rather than "suggesting the terrestrial outflows of organic matter to the ECS").

Abstract, line 31: Concentration of glucose given is given as 0.31 ng/m3. Is this concentration significant to 2nd decimal unit?

Abstract, line 32: should read as "All sugar compounds show higher (not showed) —–

Abstract, lines 33: The concept of night-time and day-time variability in concentrations is misleading. This cannot be attributed to long-range transport from continental sources. This needs to be adequately addressed in the discussion section. There are high concentrations of total sugars in day-time samples.

Abstract, line 34: English!!! —– one high-oxidation tracer——

Abstract, lines 35-36: How high abundance of only two species is attributed to transport of aged organic aerosols through long-range transport?

Abstract, lines 36-38: What are the errors associated with Fungal-spore-derived, sesquiterpene-derived and biomass-burning-derived OC?

Abstract, lines 39-41: A very obvious conclusion!!! What else is expected other than transport of continental aerosols along the cruse track over ECS? What are biogenic OCs?

Page 4, lines 117-118: English!!! "The ECS is susceptible to the outflow of continental OM ——. Oceanic regions downwind of pollution sources are influenced by the chemical species.

Page 5, Section 2.2: No details are provided for the determination of total OC in aerosol samples collected over ECS.

Page 9, lines 272-273: Based on observed correlation one cannot conclude that marine sources contribute to the particulate glucose over ECS.

Page 9, line 274: English!!! ——fructose was obviously related to glucose——

Page 10, lines 299-300: ——suggest that the major source of SOA tracers over the ECS is probably of terrestrial origin. A very obvious inference to make from measured SOA tracers.

Page 16, lines 489-490: What are the errors associated with fsoc used in the study? What is their significance to 3rd decimal units?

Page 16, lines 496-500: What is the rationale of invoking photo-degradation by free radicals in the atmosphere or other atmospheric dilution mechanism during long-range transport to the western North Pacific? Higher concentrations of biomass burning OC are not an evidence to invoke the process of photo-degradation. What is "other atmospheric dilution mechanism"?

Page 19, lines 594-597: Authors should have used $Mg^{2+}/Na^+$ ratio for deriving contribution from sea-salts due to chloride depletion in MABL. The measured $SO_4$ should be corrected for contribution from sea-salt (to use non-sea-salt $SO_4$)

Page 20, lines 605-609: Likewise, $Ca^{2+}$ should be corrected for contribution from seasalt (nss-Ca2+). The anthropogenic sulphate does react with crustal aerosols (mineral dust containing CaCO3) to form Ca2+ and SO4 ions but does not form metal sulphate CaSO4 or MgSO4. CaSO4 is a primary mineral – gypsum.

Page 21, lines 635-638: English!!! Previous study once pointed out ——. Transport pathway could be similar for NOx and BSOC.

Page 21, line 647: If the contribution from each source changes in space and time then the concentrations measured for only few days (N=21 samples) over ECS are not representative.

Page 21, lines 652-654: This is conceptually wrong. Contribution from sea-salt is not an evidence for oceanic emissions of BSOA.

Page 21, lines 654-656: What is primary and secondary OM? For the study site, continental sources are dominant is a primary objective of carrying out sampling over ECS.

Page 22, lines 662-663 and lines 670-672: These are very obvious conclusions. What else is expected other than contribution from continental sources.

---

## Author Comment (AC1) · 12 Aug 2018

We greatly appreciate your time on the review of this manuscript and thank you for the positive comments. In this paper, we analyzed the organic molecular composition of marine aerosols that were collected during a marine cruise in the East China Sea from May 18 to June 12 in 2014. Biogenic SOA tracers and sugar compounds exhibited higher levels in the samples affected by continental air masses, suggesting the terrestrial outflow of organic matter to the East China Sea. Our results demonstrate that the Asian continent can be a natural emitter of biogenic POA and SOA in the marine atmosphere, which contributes to the communities of atmospheric chemistry and

environmental studies.

---

## Author Comment (AC2) · 12 Aug 2018

Responses to Reviewer #1: (our responses are italicized in blue)

This manuscripts reports on molecular composition, abundances and spatial distribution of sugars and biogenic secondary organic aerosols in the MABL of East China Sea (ECS). A total of 51 samples were collected in May-June 2014, day-time N=25 and night-time N=26. Authors state that (page 21, line 647) the contribution of each source changes over time and varies with distance from the continent. In this context, concentrations of the measured species are not representative on temporal and spatial scale for ECS and NW Pacific. Although the reported data set provides some novelty, there are number of issues with the discussion and interpretation presented by the authors.

*Response: Thank you so much for your detailed suggestions and comments, which help us better reveal the importance and novelty of this study.*

For a general reader, authors should clearly state constituents referred as secondary organic aerosols (SOA) measured in this study. At times authors are addressing these constituents as "biogenic SOA tracers" and/or "biogenic SOA markers". This is confusing to a general reader.

*Response: SOA contains a large spectrum of organic species, which are much more than what we reported in this study. Here we mainly discussed organic molecular markers of biogenic origin, which are derived from the photooxidation of isoprene, monoterpenes and sesquiterpenes, and this is why we addressed these constituents as "biogenic SOA tracers" and/or "biogenic SOA markers". In fact, the detailed constituents of these biogenic SOA tracers have been listed in Table 1. More details can be found in lines 328-329, 411-412 and 456-458 as well.*

Title: "Characterization of biogenic primary and secondary organic aerosols——-". What primary biogenic aerosols are being referred and measured in this study? Authors have mainly discussed biogenic secondary aerosols.

*Response: Sugars, sugar alcohols, and anhydrosugars measured in this study are the primary biogenic aerosols, which were referred in lines 73-82 and lines 186-192.*

Abstract, lines 29-30: Authors state that "Biogenic SOA tracers and sugars exhibit higher levels in the samples affected by continental air masses, suggesting the terrestrial outflows of organic matter to the East China Sea". This is a redundant statement. If high concentrations of measured species are associated with continental air masses, then continental outflow of organic matter is much obvious (rather than "suggesting the terrestrial outflows of organic matter to the ECS").

*Response:* *Given that the conclusion is obvious, the word "suggesting" in the sentence has been changed to "demonstrating" in line 30.*

Abstract, line 31: Concentration of glucose given is given as 0.31 ng/m3. Is this concentration significant to 2nd decimal unit?

*Response:* *Yes, it is.*

Abstract, line 32: should read as "All sugar compounds show higher (not showed) ——

*Response:* *Changed in line 32.*

Abstract, lines 33: The concept of night-time and day-time variability in concentrations is misleading. This cannot be attributed to long-range transport from continental sources. This needs to be adequately addressed in the discussion section. There are high concentrations of total sugars in day-time samples.

*Response:* *Here we present the general variation trend observed in this study, and we did not further attribute this phenomenon to long-range transport from continental sources. The sentence "All sugar compounds showed higher concentrations in the nighttime than in the daytime" has been changed into "All sugar compounds show generally higher average concentrations in the nighttime than in the daytime" (lines 32). We also add the discussion section (lines 198-200) in the revised manuscript based on your good suggestion which makes our description more accurate and convincing.*

Abstract, line 34: English!!! —— one high-oxidation tracer——

*Response:* *"one high-oxidation tracer" has been changed to "one higher-generation photooxidation tracer" in line 34.*

Abstract, lines 35-36: How high abundance of only two species is attributed to transport of aged organic aerosols through long-range transport?

*Response:* *3-Methyl-1,2,3-butanetricarboxylic acid (MBTCA) is a high-generation photooxidation product of monoterpene. Monoterpenes were chiefly emitted from needle leaf trees. Therefore, the MBTCA' presence in large amounts indicates large input of aged aerosols, which most likely come from continent through long-range atmospheric transport.*

Abstract, lines 36-38: What are the errors associated with Fungal-spore-derived, sesquiterpene-derived and biomass-burning-derived OC?

*Response:* *Thank you for your comments. The contributions from OC derived from fungal spore, biomass burning and biogenic VOC to total OC are calculated with widely and successfully used tracer-based methods. The details can be found in lines 482-495.*

*We have added the following references to support our statement:*

*[1] Bauer, H., Claeys, M., Vermeylen, R., Schueller, E., Weinke, G., Berger, A. and Puxbaum, H.: Arabitol and mannitol as tracers for the quantification of airborne fungal spores, Atmospheric Environment, 42, 588-593, 2008.*

*[2] Fu, P., Kawamura, K., Chen, J. and Miyazaki, Y.: Secondary Production of Organic Aerosols from Biogenic VOCs over Mt. Fuji, Japan, Environmental Science & Technology, 48, 8491-8497, 2014.*

*[3] Kleindienst, T. E., Jaoui, M., Lewandowski, M., Offenberg, J. H., Lewis, C. W., Bhave, P. V. and Edney, E. O.: Estimates of the contributions of biogenic and anthropogenic hydrocarbons to secondary organic aerosol at a southeastern US location, Atmospheric Environment, 41, 8288-8300, 2007.*

*[4] Zhu, C., Kawamura, K. and Fu, P.: Seasonal variations of biogenic secondary organic aerosol tracers in Cape Hedo, Okinawa, Atmospheric Environment, 130, 113-119, 2016.*

Abstract, lines 39-41: A very obvious conclusion!!! What else is expected other than transport of continental aerosols along the cruse track over ECS? What are biogenic OCs?

***Response:*** *The influence of continental aerosols' transport on marine air is one of obvious conclusions in the study, while the contributions from marine emission to sugar compounds and biogenic SOA tracers exist as well but minor. Here we also discussed the spatial distribution of sugars and SOA tracers, as well as the major species of measured chemical components in marine aerosols over the ECS. We also found that fungal spores contribute most to total OC in comparison with secondary organic aerosols and biomass burning. PMF results identified the main sources for marine aerosols over the ECS as well, such as secondary origins and biogenic SOA.*

*Actually, biogenic OCs in this paper contain OCs derived from biomass burning, fungal spores and SOCs. In order to make this sentence to be more precise, we changed "biogenic OCs and SOCs"* **into** *"biogenic primary OCs and SOCs" (line 39-40). Thanks for your good comments.*

Page 4, lines 117-118: English!!! "The ECS is susceptible to the outflow of continental OM ——. Oceanic regions downwind of pollution sources are influenced by the chemical species.

***Response:*** *Given that the wind direction varies with the season and the ECS is not always located downwind of pollution sources, and our main focus in this study is organic aerosols, we think this sentence may be more appropriate described as "The ECS is an oceanic region susceptible to the influence from outflow of continental OM" (line 117-119).*

Page 5, Section 2.2: No details are provided for the determination of total OC in aerosol samples collected over ECS.

***Response:*** *Thank you for careful reading. The total OC in aerosol samples collected over ECS were determined using thermal optical reflectance (TOR) following the Interagency Monitoring of Protected Visual Environments (IMPROVE) protocol on a DRI Model 2001 thermal/optical carbon analyzer. The limit of detection for the carbon analysis was 0.82 $\mu gC\ cm^{-2}$ for OC. Because total OC is not our focus here, we just cite the data of total OC in aerosol samples over the ECS to estimate the contributions of biogenic primary and secondary OC in total OC (%). The detailed information on determination and more discussion of total OC as stated above will be reported in another paper which is unpublished.*

Page 9, lines 272-273: Based on observed correlation one cannot conclude that marine sources contribute to the particulate glucose over ECS.

*Response: We have carefully checked the manuscript and the reasons for this conclusion can be listed as follows: first, the short chain $C_{16:0}$ fatty acid is mainly emitted by marine phytoplankton and microbes. Our previous study also reported that short chain fatty acids in marine aerosols over the ECS are mainly derived from the ocean surface via sea spray. The good correlation between glucose and $C_{16:0}$ fatty acid in this study reflects they have similar origin. Thus, the observed correlation in this paper indicates ocean's contribution to glucose in the marine atmosphere, which is also in accordance with early report mentioned in the manuscript. This part "Medeiros and Simoneit (2007) reported that high abundance of glucose associated with lower molecular weight fatty acids (mainly $C_{16}$) was attributed to the spring bloom of algae. In our study, $C_{16:0}$ fatty acid correlated well with glucose (p = 0.001, r = 0.46, N = 51), which suggests that marine sources also contributed to the particulate glucose in the oceanic atmosphere" has been changed* **into** *"Medeiros and Simoneit (2007) reported that high abundance of glucose associated with lower molecular weight fatty acids (mainly $C_{16}$) was attributed to the spring bloom of algae. In our study, glucose correlated well with measured $C_{16:0}$ fatty acid (p = 0.001, r = 0.46, N = 51), which mainly emitted from the ocean surface via sea spray (Kang et al., 2017), suggesting that marine sources also contributed to the particulate glucose in the oceanic atmosphere" (see lines 272-276).*

Page 9, line 274: English!!! ——fructose was obviously related to glucose——

*Response: Changed to "fructose had a strong correlation with glucose" (line 276).*

Page 10, lines 299-300: —suggest that the major source of SOA tracers over the ECS is of terrestrial origin. A very obvious inference to make from measured SOA tracers.

*Response: We delete "probably" in the revision (see line 301).*

Page 16, lines 489-490: What are the errors associated with $f_{soc}$ used in the study? What is their significance to 3rd decimal units?

***Response:*** *The $f_{soc}$ values used in the study were reported by Kleindienst et al. (2007) - "Estimates of the contributions of biogenic and anthropogenic hydrocarbons to secondary organic aerosol at a southeastern US location", which were obtained using a tracer-based method and the errors associated with $f_{soc}$ were not mentioned in that paper. The significance of $f_{soc}$ is to 3rd decimal units according to the published paper by Kleindienst et al. (2007).*

*Although large errors may be raised during the estimation of SOA formation using laboratory derived factors, these $f_{soc}$ values have been widely and successfully used to identify major sources of SOA in a wider range of previous studies.*

Page 16, lines 496-500: What is the rationale of invoking photo-degradation by free radicals in the atmosphere or other atmospheric dilution mechanism during long-range transport to the western North Pacific? Higher concentrations of biomass burning OC are not an evidence to invoke the process of photo-degradation. What is "other atmospheric dilution mechanism"?

***Response:*** *What we actually want to express is as follows: biomass burning OC showed higher levels in the samples near the Asian continent than those over the remote ocean. Hence, we deduce that these continental biomass-burning OC must have undergone removal or degradation during the long-range transport to ocean. For example, levoglucosan, a key biomass burning tracer, can be degraded by free radicals like OH in the atmosphere reducing its atmospheric concentration during long-range transport. "Other atmospheric dilution mechanism" can be some atmospheric mixing processes which also decrease the atmospheric concentration of biomass burning OC apart from wet/dry deposition.*

Page 19, lines 594-597: Authors should have used Mg2+/Na+ ratio for deriving contribution from sea-salts due to chloride depletion in MABL. The measured SO4 should be corrected for contribution from sea-salt (to use non-sea-salt SO4)

***Response:*** *Thanks for your suggestions. We think that $Na^+$ may be more appropriate for estimating marine contribution, for $Na^+$ in particulate samples originates solely from sea salt through sea spray and is widely used as a marker for marine emission. While $Mg^{2+}$ is present in crustal dust in large quantities.*

*The source apportionment with PMF in previous studies mostly used measured $SO_4^{2-}$ as variables. And the input data of PMF contains the observed data and the corresponding uncertainties matrix. The calculation of uncertainties needs to use method detection limit (MDL), and the detailed methods and equation were listed in lines 160-168 in the manuscript. If we use non-sea-salt $SO_4^{2-}$, we cannot obtain its MDL directly. And even though we finally estimate the MDL of non-sea-salt $SO_4^{2-}$ through some way, the estimation errors would be definitely increased. So, we chose measured $SO_4^{2-}$ as a variable for PMF in accordance with extensive early reports.*

Page 20, lines 605-609: Likewise, Ca2+ should be corrected for contribution from sea-salt (nss-Ca2+). The anthropogenic sulphate does react with crustal aerosols (mineral dust containing CaCO3) to form Ca2+ and SO4 ions but does not form metal sulphate CaSO4 or MgSO4. CaSO4 is a primary mineral – gypsum.

**Response:** *The reason to use $Ca^{2+}$ rather than nss-$Ca^{2+}$ as a variable for PMF is the same as $SO_4^{2-}$ as stated above.*
*Thanks very much for your careful reading and good suggestions. Considering that discussion of sulphate here is not that necessary for crustal sources, we deleted this part in order to make the whole manuscript clearer and more concise to future readers.*

Page 21, lines 635-638: English!!! Previous study once pointed out ——. Transport pathway could be similar for NOx and BSOC.

**Response:** *We have changed "Previous study once pointed out" **into** "Previous studies have reported that" in line 636 based on your suggestion. And the former sentence is also rephrased as "$NO^{3-}$ and BSOA may share common formation and/or transport pathways" (line 635-636) as you suggested.*

Page 21, line 647: If the contribution from each source changes in space and time then the concentrations measured for only few days (N=21 samples) over ECS are not representative.

**Response:** *Thanks for your careful comments. The marine aerosol samples were collected from May 18 to June 12, 2014. Actually, samples for 26 days were obtained, and totally fifty-one daytime and nighttime samples were achieved with some additional blank ones. As we all know, marine aerosol samples are very valuable due to the*

*difficulty in sampling. The samples used in this study were obtained during a National Natural Science Foundation of China (NSFC) sharing cruise. Due to limited cruise period, we try our best to obtain as many samples as possible, and that is one reason that we sampled during both the day and the night. In order to ensure the accuracy of observation and discussion, we've done plenty of work and compared with other reports about marine aerosols. In addition, we hope there will be another golden opportunity to sample more marine aerosols in the future, so we can make further study and obtain more plentiful observation data.*

Page 21, lines 652-654: This is conceptually wrong. Contribution from sea-salt is not an evidence for oceanic emissions of BSOA.

***Response:*** *Sea-salt sources here refer organic matter emitted from the ocean surface together with sea-salt particles via sea spray or bubble bursting. We have rephrased this sentence in the revised manuscript (lines 657-659) to be clearer.*

Page 21, lines 654-656: What is primary and secondary OM? For the study site, continental sources are dominant is a primary objective of carrying out sampling over ECS.

***Response:*** *The primary OM refers OM derived from biomass burning, fungal spores, pollens and plant debris, and some other primary sources which can emit OM directly into the atmosphere. Secondary OM in the present study refers photooxidation products of biogenic VOCs.*

*During this cruise sampling, we sampled marine aerosols both over the remote sea and close to Asian continent and make a lot of comparison between them. The obvious difference in chemical species illustrates the crucial role of continental sources in the marine atmosphere.*

Page 22, lines 662-663 and lines 670-672: These are very obvious conclusions. What else is expected other than contribution from continental sources.

***Response:*** *In fact, we also measured the chemical composition, abundance and major sources of sugar compounds and biogenic SOA tracers in marine aerosols, then identified the dominant species and their spatial distribution as well as diurnal variation in the ECS, providing basic observational data for studying marine aerosols.*

*In order to compare the relative contribution of marine and continental sources to total OC in the oceanic atmosphere, the whole sampling area was also divided into five regions from north to south according to the spatial distribution of aerosol samples. According to our current knowledge, reports on biogenic SOA tracers are still limited so far, so this work is very meaningful for better understanding the climatic impacts and biogeochemistry of marine aerosols.*

Other modifications in the revised manuscript:

*We replaced Figure 11 with another pie graph that illustrates contributions of different sources to organic carbon (OC) in marine aerosols over the East China Sea, because this study mainly focuses on organic aerosols in the marine atmosphere. The new Figure 11 is directly pertinent to the themes of organic aerosols. We also added some discussion in the main text to clarify it (line 645-649), and rephrased some sentences regarding new Figure 11 in the Abstract (line 41-43) and Conclusions (line 676-677) section to make it more clear.*